# On deep convection events and Antarctic Bottom Water formation in ocean reanalysis products

Wilton Aguiar[1], Mauricio M. Mata[1] and Rodrigo Kerr[1]

[1]Laboratório de Estudos dos Oceanos e Clima, Instituto de Oceanografia, Universidade Federal do Rio Grande – FURG. Rio Grande, RS, 96203-900, Brazil.

*Correspondence to*: Wilton Aguiar (aguiar.wilton@gmail.com)

**Abstract**

Open ocean deep convection is a common source of error in the representation of Antarctic Bottom Water (AABW) formation in ocean general circulation models. Although those events are well described in non-assimilatory ocean simulations, the recent appearance of a massive open-ocean polynya in the Estimating the Circulation and Climate of the Ocean Phase II reanalysis product (ECCO2) raises questions on which mechanisms are responsible for those spurious events and if they are also present in other state-of-the-art assimilatory reanalysis products. To investigate this issue, we evaluate how three recently released high-resolution ocean reanalysis products form AABW in their simulations. We found that two of the products create AABW by open ocean deep convection events in the Weddell Sea that are triggered by the interaction of sea ice with the Warm Deep Water, which shows that the assimilation of sea ice is not enough to avoid the appearance of open ocean polynyas. The third reanalysis, My Ocean University Reading UR025.4, creates AABW using a rather dynamically accurate mechanism. The UR025.4 product depicts both continental shelf convection and the export of Dense Shelf Water to the open ocean. Although the accuracy of the AABW formation in this reanalysis product represents an advancement in the representation of the Southern Ocean dynamics, the differences between the real and simulated processes suggest that substantial improvements in the ocean reanalysis products are still needed to accurately represent AABW formation.

## 1 Introduction

Recently, different groups of experts have developed several state-of-the-art eddy-permitting general ocean circulation models with long simulations, elegant and efficient assimilation methods. Based on those models, ocean reanalysis products, which reconstruct oceanic features using governing ocean equations and observed data, have been coupled with global climate models (GCMs) to produce detailed climate estimates (Lee et al., 2009). Specific climate-induced studies using ocean reanalysis products focus on several features, such as descriptions of source water mass

contributions (Kerr et al., 2012a), estimates of sea level variability (Berge-Nguyen et al., 2008; Köhl and Stammer, 2008; Wunsch et al., 2007), evaluations of surface circulation and the heat content of specific ocean basins (Schiller et al., 2008; Zhu et al., 2012), and description of the decadal variability of ocean heat content (Carton et al., 2005). One of the features receiving special attention in ocean reanalysis products is the representation of the lower limb of the Atlantic Meridional

Overturning Circulation (AMOC). In this sense, recent assessments have revealed several model inconsistencies related to the Southern Ocean dense water formation and export, which is a key process in the AMOC lower limb dynamics (e.g., Azaneu et al., 2014).

The dense bottom waters in the Southern Ocean are mainly formed by two mechanisms. The first mechanism is through (i) a complex interaction of deep and shelf waters and starts with deep waters, originally formed in the North

Atlantic, being transported to the south through the AMOC. During transport, the deep water changes properties and along the way forms Circumpolar Deep Water (CDW) as it enters the Southern Ocean (e.g., Talley, 2013). There, the CDW circulates along with the Antarctic Circumpolar Current (ACC) and is eventually advected towards the Antarctic coastal margin. Near the coastal margins, the interaction of CDW-derived waters with High Salinity Shelf Waters (HSSW), which are formed from brine released during the winter, enhances the CDW density and creates Antarctic Bottom Water (AABW;

Carmack and Foster, 1975; Foster and Carmack, 1976). Alternatively, HSSW can circulate under ice shelves, losing heat and salt to create Ice Shelf Water. Ice Shelf Water then flows downslope and mixes with deep waters to create AABW (Foldvik et al., 1985). The coastal AABW formation occurs primarily in the Weddell Sea, which is considered one of the zones in the Southern Ocean with the highest AABW production (e.g., Orsi et al., 1999; Kerr et al., 2012b). A few other regions around Antarctica also contribute to bottom water formation, such as Prydz Bay (e.g., Williams et al., 2016), Adélie Land (e.g.,

Williams et al., 2008) and the Ross Sea (e.g., Whitworth and Orsi, 2006).

This complex coastal bottom water formation process is notably difficult to represent in GCMs, and GCMs instead create AABW through an alternative mechanism of (ii) open ocean deep convection. Open ocean deep convection in the Southern Ocean occurs when the water column stability decreases allowing heat transference to the surface. This heat transfer creates an ice-free region, which by definition is an open ocean polynya. In the open ocean polynya, saline deep

waters lose sensible heat to the atmosphere, creating AABW by cooling (Killworth, 1983). This process rarely occurs in the real ocean. In fact, the last known large open ocean polynya event was documented by Gordon (1978) and Carsey, (1980) who reported a feature 350,000 km$^2$ in size during the winters of 1974-1976 in the Weddell Sea. Although smaller ocean polynyas occurred in the 20$^{th}$ century (Comiso and Gordon, 1987), no winter ice-free areas in the Southern Ocean with the dimensions and persistence of the Weddell Polynya have been reported since the 1970s.

Nevertheless, ocean simulations recurrently represent bottom water formation by spurious open ocean deep convection events. Recently, Azaneu et al. (2014) evaluated the AABW properties in the Estimating the Circulation and Climate of the Ocean Phase II (ECCO2) and found that an intense pulse of AABW formation occurs in this reanalysis as a result of the opening of an unrealistic polynya in the Weddell Sea. After the polynya opens, the bottom layer transports and densities become unrealistically high, and all Southern Ocean representations become unreliable. Additionally, Heuzé et al.

(2013) found that most models of the Coupled Model Intercomparison Project (CMIP) Phase 5 failed to represent the formations of dense waters accurately and instead created AABW by open ocean deep convection. Several other simulations have also reported the creation of AABW from unexpected open ocean polynyas (Marsland et al., 2003; Timmermann and Beckmann, 2004; Shaffrey et al., 2009). The frequent occurrence of open ocean deep convection events in simulations raises a need to understand if those unrealistic polynyas are found in the recently released reanalysis products, what is their opening mechanism and how those products without open ocean polynyas represent the AABW formation issue. In this study, we investigated three recent ocean reanalysis products with documented evidence of AABW formation to determine if open ocean convection is the most common reason for anomalous AABW formation in the assimilatory GCMs. Moreover, we investigate the other mechanisms by which bottom water is created in each ocean reanalysis.

## 2.1 Ocean Reanalysis Datasets and Observations

Three ocean reanalysis products were evaluated in this study. The first product investigated was ECCO2, which was chosen to be used as a comparison standard to the other reanalysis products due to the reported deep convection event triggered by the polynya opening in the Weddell Sea after 2003 (Azaneu et al., 2014). This coupled ocean reanalysis is based on the Massachusetts Institute of Technology General Circulation Model (MITgcm) with a cube-sphere grid, as described by Marshall et al. (1997). The cube-92 solution used of ECCO2 is forced by the atmospheric Japanese 25-Year Reanalysis (JRA-25). A sea ice model by Zhang et al. (1998) that estimates snow cover, sea ice thickness and concentration is incorporated in the ECCO2 framework. The data assimilated by ECCO2 includes temperature and salinity profiles from the World Ocean Circulation Experiment database, Argo floats, and XBT measurements; sea surface temperature measurements from the Group of High Resolution Sea Surface Temperature (GHRSST); sea level anomaly data from altimetry; temporal mean sea levels from Maximenko and Niiler (2005); sea ice concentrations from passive microwave data; sea ice thickness from Upward Looking Sonar; and finally sea ice motion from the QuikSCAT and RADARSAT-GPS radiometers. A Green's function method is used to calibrate the control variables (Menemenlis et al., 2005) and the initial parameters, which include initial temperature and salinity conditions; background vertical diffusivity; atmospheric surface boundary conditions; critical Richardson numbers; air-ocean, ice-ocean and air-ice drag coefficients; albedo coefficients of ice, ocean and snow; bottom drag and vertical viscosity. ECCO2 is run directly from its initial conditions, without the use of a spin-up period to bring the model to equilibrium (Aksenov et al., 2016). The ECCO2 reanalysis product spans from 1992 to 2012, with a 0.25°x0.25° horizontal resolution and 50 unevenly spaced vertical levels (Menemenlis et al., 2008).

We chose to work with two other reanalysis products that had evidence of rapid AABW formation, i.e., rapid density increases in deep and bottom waters: *Southern Ocean State Estimate version 2* (hereafter referred to as SoSE) and *My Ocean University of Reading* (hereafter referred to as UR025.4). The UR025.4 exhibits increasing neutral densities in both the deep and bottom layers of the Weddell Sea after 2004 (Dotto et al., 2014), which suggests AABW formation. UR025.4 uses the NEMO version 3.2 ocean circulation model, which is forced by the ERA-Interim Atmospheric Reanalysis and incorporates Louvain-la-Neuve Ice model Version 2 (LIM2;Fichefet & Maqueda, 1997). The Optimal Interpolation

scheme (Storkey et al., 2010) from the UK Met Office operational FOAM-NEMO system was used to assimilate the ocean variables. UR025.4 spans from 1993 until 2010 and has a tripolar grid with a mean horizontal resolution of 0.25°x0.25° and 75 vertical levels (Ferry et al., 2012). The assimilated UR025.4 data includes temperature and salinity profiles from the EN3 dataset, including Argo floats, XBT, CTD, TAO and PIRATA measurements; sea surface temperature and altimetry data from the International Comprehensive Ocean-Atmosphere Data Set; and sea ice concentration from the Ocean and Sea Ice Satellite Application Facility. UR025.4 uses initial conditions of EN3 climatology to start the simulation. Authors considered that the 3d-Assimilation scheme allowed fast adjustment of surface and subsurface properties, and hence no spin up period is used in this reanalysis (Valdivieso et al., 2014).

The SoSE reanalysis have documented wintertime deep convection during the test runs (Mazloff et al., 2010). The SoSE also uses the MITgcm ocean model, but it estimates the air-sea buoyancy fluxes using the NCEP-National Center for Atmospheric Research reanalysis 1 as an initial guess of the atmospheric state. The framework includes a sea ice model by Hibler (1980) and assimilates through a least squares fit with observations to reduce model error. The data constraints of SoSE include temperature and salinity fields from Argo floats and instrument-mounted elephant seal profiles; CTD and XBT profiles from the Scripps Institution of Oceanography High Resolution CTD/XBT network and the CliVar and Carbon Hydrographic Data Office; sea surface height from the Radar Altimetry Database System; sea surface temperature from microwave radiometers; sea ice concentrations from the National Snow and Ice Data Center; mean dynamical topography from the Technical University of Denmark; and bottom pressure estimates from the ECCO project. The other measurements used in the assimilations were taken from the Antarctic Marine Living Resources Program, the Long-Term Ecological Research Network and the Diapycnal and Isopycnal Mixing Experiment in the Southern Ocean. The SoSE initialization includes a one-year spin up period using the dataset from the 2004 Ocean Comprehensible Atlas (OCCA - Forget, 2010) with adjusted kinetic energy. The optimization method applied in the SoSE changes the initial temperatures and salinities, and a one-week adjustment shock occurs when the model begins to run. Furthermore, neither the OCCA nor the SoSE were optimized to eliminate spurious drifts (M. Mazloff, personal communication). The SoSE has a horizontal resolution of 0.16°X0.16°, 42 irregular vertical levels, and a time span from 2005 until 2010. Finally, the distinct simulation characteristics between the three reanalysis products, such as the initialization methods and the assimilated variables, help track how the different features in the simulation frameworks affect AABW production.

As sea ice formation and melting have direct connections to AABW formation, the mean sea ice concentration (SIC) and thickness (SIT) have been analyzed in the present study. ECCO2 and SoSE are available on the National Aeronautics and Space Administration Jet Propulsion Laboratory (NASA; http://ecco2.jpl.nasa.gov/) and Scripps Institute of Oceanography (http://sose.ucsd.edu/) websites, respectively. The UR025.4 simulations are available on the Centre for Environmental Data Analysis of the United Kingdom website (http://catalogue.ceda.ac.uk/uuid/ef3e53aef4dca2030ebc9e84aa908d74).

For a better description of the distinct regional AABW formation processes and sea ice patterns, we split the Southern Ocean into five sectors (Figure 1) following Parkinson & Cavalieri (2012) :130°W to 60°W is the Bellingshausen

and Amundsen Seas sector, 60°W to 20°E is the Weddell Sea sector, 20°E to 90°E is the Indian Ocean sector, 90°E to 160°E is the Western Pacific sector and 160°E to 130°W is the Ross Sea sector. The annual averages of the sea ice concentration and thickness were compared by sector between the reanalysis products to identify the relations of the ice with the AABW formation processes. A comparison with an observational dataset was necessary to grasp the veracity of the sea ice alterations, and the sea ice concentration product derived from the Special Sensor Microwave/Imagers (SSM/I) and the Special Sensor Microwave Imager/Sounder (SSMI/S) provided by the National Snow and Ice Data Center was used (https://nsidc.org). Both SSM/I and SSMI/S originated from the Nimbus-7 Scanning Multichannel Microwave Radiometer.

## 2.2 Methods

We analyzed the oceanic regions south of 60°S and estimated the volume of water masses by sector. To calculate the water mass volumes, we used the water mass definitions by neutral density layers ($\gamma^n$; Jackett & McDougall, 1997; Serazin, 2011). The three reanalysis products provide potential temperature and salinity, which were used to calculate the neutral density throughout this study. For the majority of the sectors, three water masses were defined, as described in Table 1: AABW, Lower Circumpolar Deep Water (LCDW), and Upper Circumpolar Deep Water (UCDW), with neutral density limits following Abernathey et al. (2016). The waters with densities lower than 27.7 kg m$^{-3}$ were analyzed as a single group of "surface waters". As the Weddell Sea has a unique water mass structure (Orsi et al., 1999), the layers were split into Weddell Sea Bottom Water (WSBW), Weddell Sea Deep Water (WSDW), Warm Deep Water (WDW) and Antarctic Surface Water (AASW) in the shallowest end (Naveira Garabato et al., 2002; Franco et al., 2007). The total volume of each water mass by sector was calculated by an integration described in Eq.(1):

$$V_{wm} = \frac{100}{V_{sector}} \int_{L_{coast}}^{60°S} \int_{Wl}^{EL} \int_{z_{\gamma 2}}^{z_{\gamma 1}} dz \, dx \, dy \text{, (1)}$$

Where $z_{\gamma 1}$ and $z_{\gamma 2}$ are the depths of the upper and lower neutral density limits of each water mass in each cell, respectively, and $dx$ and $dy$ are the zonal and meridional lengths of each cell, respectively. Although $dx$ and $dy$ are not constant throughout the reanalysis grids, the integration process takes that into account, so the water mass estimates were not contaminated by errors due to the non-uniform cell size. The result of the vertical integration is then integrated meridionally between 60°S and the latitude of the coastline ($L_{coast}$) and zonally between the eastern (EL) and western (WL) limits of each sector. Thus, the result of Eq. (1) is the percentage of the volume of the water mass ($V_{wm}$) relative to the total water volume of each sector ($V_{sector}$). Those monthly volumetric percentages were then used to infer the transformation and pulses of AABW as well as the processes involved.

Changes in salinity and temperature in the Southern Ocean can be used as proxies to determine brine release, surface cooling and water mass entrainment during AABW formation. Hence, the time series of those hydrographic

properties were analyzed to discuss the presence of the abovementioned processes during bottom water formation. The averaged anomalies, which considered the long-term average of both temperature and salinity, were calculated for all ocean reanalysis products in three distinct layers: (i) a surface layer from 100 m to 150 m, (ii) an intermediate layer from 400 m to 650 m, and (iii) a bottom layer from 3000 m to the reanalysis seafloor. The depth limits of the three layers were chosen

specifically due to their links with the processes being evaluated (Orsi et al., 1999). The surface layer limits were chosen because their depths record the temperature and salinity signals related to brine release and surface cooling; the intermediate limits are consistent with the depths of the deep waters and hence record the changes in the deep water properties, and the bottom layer mainly records changes in the AABW, which constitutes most of the Southern Ocean below 3000 m. The temperature and salinity anomalies were calculated for the Indian Ocean sector, the Western Pacific sector and the Weddell

Sea sector due to the presence of AABW formation in those locations. Additionally, to make the time series of the variables comparable between the sectors, the data were normalized by their standard deviations since the inherent salinity and temperature of each layer is different for each sector. Those standardized anomalies provide an estimate on how much the temperature and salinity deviates from the long-term average, which is a useful approach to identify the low-frequency salinity and temperature changes in the time series, such as the ones related to AABW formation. The annual and monthly

time series of the standardized anomalies in each sector were analyzed, while focusing on the period and location of AABW formation to help explain the mechanisms involved in the formation. Finally, for a better description of the AABW formation process in UR025.4, we included analyses of the sea ice and ocean currents, all of which were provided by the reanalysis product.

## 3 Results and Discussion

20       In this section, we first describe the average sea ice patterns in the Southern Ocean sector, its spatial signature and evidence that this property is related to the AABW formation in the reanalysis products investigated (Sect. 3.1). Section 3.2 discusses the water mass volume transformations in each sector, and attempts to identify the AABW formation in the different products (Sec. 3.2). In section 3.3, the salinity and temperature anomalies along the water column were explored during the periods of AABW formation to identify the roles of brine release events, surface cooling and water mass changes

in AABW formation in each reanalysis product. Finally, in section 3.4, we explain how the mechanisms of AABW formation are occurring in the three reanalysis products, and discuss the impact of AABW formation in each simulation.

### 3.1 Sea Ice Concentration and Thickness

All three reanalysis products overestimate the annual mean SIC compared to the National Snow and Ice Data Center observations (hereafter referred to as NSIDC) in all sectors until 2004, except in the Ross and Bellingshausen Seas, where

the concentrations are similar to the observations obtained from the NSIDC (Figure 2a-f). The highest overestimates occur in

the Weddell and Indian Ocean sectors, where both the ECCO2 and UR025.4 cells predict at least 20% more SIC than the observations (Figure 2b,e). Previous experiments with LIM2, under the atmospheric forcing of NCEP/NCAR, show enhanced sea ice extent within the seasonal cycle and a tendency to overestimate the SIT in the Weddell Sea. The pattern of SIT overestimation in the previous LIM2 study was reportedly due to the westerlies in the NCEP/NCAR forcing being stronger than reality (Massonnet et al., 2011). Although UR025.4 uses the LIM2 model, an atmospheric forcing different from the previous study is applied in this study (ERA-INTERIM). Careful analyses of the coastal wind speeds and directions reveal no significant overestimation of the westerlies. Moreover, previous validation of the ERA-INTERIM wind fields have not reported overestimation of the westerlies around the Antarctic Peninsula (Dee et al., 2011). Nevertheless, UR025.4 still overestimates the SIT in the Weddell Sea (Figure 2b).

The analysis performed by Azaneu et al. (2014) showed that the high annual SIC in ECCO2 is due to an overestimation of the maximum sea ice in the winter, which raises the annual mean. However, ECCO2 is the reanalysis product that best represents the SIC in the time series before 2004 (Figure 2a). It is important to highlight that all reanalysis products examined in this study exhibit interannual variability patterns that are very close to observations. The most noticeable spurious signals in sea ice only occur after 2004. The SIC and SIT in the Weddell Sea and Indian Ocean sectors in ECCO2 decrease quickly by approximately 20%, hence lowering the whole Southern Ocean average (Figure 2a,b,e). This decreasing pattern is unrealistic since satellite observations from the NSIDC show no such alterations. Additionally, the estimates of sea ice extent in those sectors of real Southern Ocean point to an increase until 2010 (Parkinson & Cavalieri 2012). Hence, the anomalous sea ice content in the ECCO2 hints to the appearance of an oceanic polynya. Conversely, UR025.4 exhibits annual SIT increases in the Weddell Sea, almost doubling that signaling 2009 (Figure 2b). Although no observational SIT database that efficiently covers the Southern Ocean is currently available to our knowledge, the comparisons among the three reanalysis thicknesses and the magnitudes of the signals suggest an UR025.4 overestimate, especially in Weddell Sea. Such sea ice thickening events have the potential to create AABW (due to increased brine rejection). Despite the availability of SoSE sea ice concentration and thickness values only between 2005 and 2010, their variabilities seem to follow the observations, with mean SIC being also higher than the observed (Figure 2a-f). The SoSE has annual mean SIT values close to ECCO2 (Figure 2g-l). Since the annual values of SIC and SIT from the SoSE vary smoothly, significantly high pulses of AABW are not expected to occur in this reanalysis.

The anomalous sea ice patterns in the ECCO2 and UR025.4 reanalysis products point to AABW formation by polynya opening and increased brine release, respectively in the Weddell Sea sector. Further analysis of the water mass contents in each model is performed to investigate this issue.

**3.2 Water Mass Percentages by Sector**

The water mass percentages from both the ECCO2 and UR025.4 reanalysis products of all water masses in the Weddell Sea sector are very similar at the beginning of the time series. However, the ECCO2 product exhibits the highest

shift in the percentages of water masses from the beginning of the time series to the end (Figure 3). AASW occupies 10% of the Weddell Sea sector in the ECCO2 reanalysis product in 1992, and its contribution decreases throughout the time series. An initially slow decrease occurs until 2000, and after that, the percentage declines at a higher rate and reaches less than 5% at the end of the series, i.e., half of its initial volume. With the decrease of the surface water volume, the seasonal cycles of

AASW formation seem more apparent (Figure 3 –AASW). While the AASW percentages decline, the WDW volume appears to increase from its initial value of approximately 36% until 2004 (Figure 3 – WDW). Pardo et al. (2012) evaluated the mean volume of deep and bottom waters below 45°S and found that the Weddell Sea water column was comprised of approximately 25±8% of NADW. Within the Weddell Sea, NADW is transformed, and part of it becomes WDW after entering the Weddell Gyre (Carmack, 1974); hence, the 36% value of WDW in ECCO2 is an overestimation, because it

surpasses the total percentage of its more widely distributed source water (NADW). The slow and steady increase of WDW content shown in Figure 3 overestimates the WDW content even more in the Weddell Sea sector, reaching its highest value of 38% in 2000 and remaining relatively stable until 2004. After that, the volume percentages of WDW swiftly decay. This change leads the decrease of WDW to volume percentages lower than 10% in 2012 (Figure 3 – WDW). The initial reduction of AASW volume and increase of the underlying WDW volume causes the core of WDW to migrate to progressively

shallower depths, which enhances its mixing with the surface. The WDW is warmer and saltier than AASW, so intensive winter surface cooling of this water mass due to polynya opening from November of 2003 until the end of the reanalysis period has the potential to form WSDW and even WSBW through open ocean convection. In fact, after 2000, when WDW reaches its highest percentage, the WSDW volume begins to gradually increase, and AABW production begins (Figure 3 – WDW and WSDW). After 2004, WSDW formation becomes more intense and its volume percentages increase sharply by

10% during the following 4 years. After 2008, it appears that WSBW begins to form in the Weddell Sea during the winters, a formation process that persists until the end of the time series. During the WSBW formation, WSDW is no longer formed, and rapid conversion of 42% of WSDW to WSBW occurs. In fact, during this period, the WSBW percentages rise from 9% to an unrealistic 70% of the water volume in the Weddell Sea (Figure 3 – WSBW).

        In UR025.4, the water mass changes in the Weddell Sea sector induce small amplitude oscillations in WSDW and

WSBW (Figure 3). The AASW volume in UR025.4 has more pronounced seasonality from 1994 until the winter of 2005 than in ECCO2. In the winter of 2005, a 2.5% drop in AASW occurs, and another 3% drop is evident in 2008. Thus, the total volume of AASW declines in total by 5.5% throughout the time series (Figure 3 – AASW). During consecutive winters, the WDW volumes drop by 2 to 6 %, whereas the percentages of WSDW and WSBW slightly increase by the same total percentage volume. This opposite pattern also shows a seasonal tendency of conversion of AASW and WDW into WSDW

and WSBW. Although the WSDW and WSBW formations in the Weddell Sea predicted by UR025.4 are lower than ECCO2, two distinct pulses of WSBW occur in the winter of 2008 (3.3%) and 2009 (6%). These events are probably due to the input of salt that occurs during the SIT increase episodes (Figure 3 – WSBW).

        The SoSE reanalysis product shows similar water mass alterations to that of the ECCO2 product prior to 2008. Although the SoSE time series is shorter, it is easy to see that the WDW volume has its peak value from January to May of

2005. From May until November of 2005, i.e. while an open ocean polynya is open in Weddell Sea, the relatively high-water volume of 32.6% of WDW swiftly decreases to 26% (Figure 3 - WDW). During that period, the WSDW percentage rises by 6%, pointing to the transformation of WDW into denser WSDW during the winter (Figure 3 – WSDW). Moreover, WSBW is also formed in the beginning of the winter of 2005. A small increase of WSBW from 14.8% to 16.6% occurs at the

beginning of the winter of 2005, but by the end of the winter, the WSBW volume returns to 14.9%, thus showing no net conversion (Figure 3 – WSBW). In the following winters of 2006 and 2007, the WSDW created in 2005 keeps being converted to WSBW, as evidenced by the WSBW pulses (Figure 3 – WSBW). In fact, during the winters of 2006 and 2007, the WSDW percentage decreases match the rise in WSBW and AASW values. As observed in ECCO2, the process seems to initiate with a high WDW content, and this water masses interacting near the surface in the Weddell Sea (Figure 3 – WDW).

After 2008, SoSE do not show either WSDW or WSBW formation (Figure 3 – WSDW and WSBW), while both AASW and WDW increase steadily by less than 5% from 2008 until 2010.

The ECCO2 water mass volumes in the Indian Ocean sector exhibit similar changes as those reported for the Weddell Sea sector, but with different timings. In this sector, the water mass distributions along the water column are different from the Weddell Sea distributions, so the analysis was carried with the appropriate vertical layers (as explained in

Sect. 2). The contents of the deep waters also seem to decrease over the time. Specifically, the UCDW decreases from approximately 4% in the winter of 2005 to almost ~0% in the winters of 2010-2012. The LCDW shows a sharper decrease in volume, from 53% in June of 2005 to 18% in June of 2012 (Figure 4 – UCDW and LCDW). Although the contents of the deep waters decrease in ECCO2, the AABW content increases from 43% to 82% in the sector, which provides evidence for the conversion of LCDW and UCDW to AABW (Figure 4 – AABW). Different from the ECCO2, the highest pulse of

AABW formation in UR025.4 originates from the Indian Ocean sector. A pulse of AABW occurs during the winter of 2004 (Figure 4 - AABW) with values spanning from 43.4% to 51.8%. During the same winter season, the LCDW volume decreases by 6% (Figure 4 – LCDW). After that winter, the AABW volume remains high, and the volume of LCDW remains low until the end of the analyzed period. That pattern provides evidence that the LCDW was transformed to AABW. The 2% remaining after the conversion is a product of the UCDW. In May-2004, the volume of UCDW increases by 3%, followed

by a 5.5% decrease, which results in a net decrease of 2.5% in UCDW volume by September 2004 (Figure 4 – UCDW). This net decrease complements the observed AABW volume increase, i.e., AABW is formed by the conversion of 2.5% of the UCDW and 6% of the LCDW. In the SoSE reanalysis, no pulse of AABW is observed, indicating that no significant volumes of AABW are formed or transformed in this sector of this model (Figure 4 – AABW).

The temporal series of the water masses from the ECCO2 reanalysis for the Western Pacific sector shows a similar

pattern as seen in the Indian Ocean sector. The UCDW volume decreases from 7% in June of 2005 to 2% in June of 2012, while LCDW in 2005 fills 80% of the water column and decreases to 50% of the water column in 2012 (Figure 5 – UCDW and LCDW). Although the contents of the deep waters decrease, the AABW volume increases from 11% in June of 2005 to 43% in June of 2012 (Figure 5 – AABW). The remaining 3% of the conversion comes from the surface waters (Figure 5 – Surface). Tomczak and Liefrink (2005) analyzed the mean AABW contribution in the Western Pacific sector using ocean

observations from the SR03 World Ocean Circulation Experiment transect (between 130°E and 150°E, and from 44°S to 66°S). The study found that AABW fills approximately 30% of the sector, a percentage lower than the 43% found in ECCO2 in 2012. The ECCO2 signature of AABW production, i.e., decreasing deep waters and increasing AABW, is seen in all sectors except the Bellingshausen and Amundsen Seas (Figures 5 -7). A similar process to the one revealed by the investigation of the UR025.4 Indian Ocean sector occurs in the Western Pacific sector. An AABW pulse with a 10% increase in volume occurs (Figure 5 – AABW) simultaneously with a total decrease of 8% in LCDW and UCDW volume in the winter of 2004 (Figure 5- UCDW and LCDW). The remaining 2% is converted from the surface water volume. Those percentage alterations also show a conversion of CDW to AABW. However, different from the Indian Ocean sector, no previous rise in UCDW volume is clear in the volumetric percentages. The SoSE AABW volume percentages show no pulses of AABW formation in the Western Pacific or any of the remaining sectors (Figure 5-7 – AABW).

The Ross Sea water mass time series does not show major AABW formation in ECCO2 until 2010 (Figure 6 - AABW). In the first eighteen years of the reanalysis (1992-2010), the LCDW volume percentage rises, while the AABW volume percentage decreases (Figure 6 – LCDW and AABW). After 2010, AABW in ECCO2 shows a small increase in volume. A slight pulse of AABW formation is also seen in UR025.4 during the winter of 2010 and before the winter of 2003, while no AABW formation is noticeable in SoSE (Figure 6 – AABW). Finally, no pulse of AABW is seen in any of the three reanalysis in the Bellingshausen and Amundsen sector (Figure 7 - AABW), which is expected since this sector in the Southern Ocean lacks hydrographic, shelf morphology and cryosphere conditions required to create AABW varieties (Potter and Paren, 1985; Orsi et al., 1999).

### 3.3 Temperature and Salinity Anomalies

The investigation of the temperature and salinity time series focus on the periods and locations of identified AABW formation in each reanalysis output, which in ECCO2 is from 2004 until 2012 at Weddell Sea, Indian Ocean and Western Pacific Sectors; in SoSE is at Weddell Sea in 2005; and in UR025.4 at Indian Ocean and Western Pacific Sectors in 2004 and Weddell Sea Sector in 2008. In ECCO2, the temperature anomaly time series shows two major trends (Figure 8). From 1992 to 2008, the intermediate layers of all sectors cool slightly (Figure 8b), and the bottom layers of all sector warm (Figure 8b-c), especially in the Western Pacific and Indian Ocean sectors. After 2008, an intense cooling is present in the intermediate and bottom layers of ECCO2, and it persists until the end of the reanalysis (Figure 8b-c). The surface layer also experiences cooling from 2008 until the end of the time series (Figure 8a). The anomalies in the ECCO2 bottom layer also show a salinity increase from 1992 to 2004 in the Weddell Sea and Western Pacific sectors, while the Indian Ocean sector does not show any clear trend before 2004 (Figure 8f). The intermediate layer of the Weddell Sea sector seems to increase in salinity throughout the entire time series, going from a –1 anomaly unit in 1992 to 2 anomaly units in 2012, while the Western Pacific and Indian Ocean sector salinities oscillate interannualy (Figure 8e). The surface layer also continuously

increases in salinity from 1992 to 2012 (Figure 8d). After 2006, the salinities strongly decrease in the bottom layer of all sectors analyzed (Figure 8f).

The intense cooling and freshening in the ECCO2 bottom layers in the Western Pacific, Indian Ocean and Weddell Sea sectors after 2006 indicate that the AABW formed in these sectors is characterized by low temperatures and salinities (Figure 8c,f). Since freshening lowers water mass densities, cooling might be one mechanism responsible for AABW formation in ECCO2. This period of bottom layer cooling coincides with the lowest annual SIC and SIT in the Weddell Sea and the Indian Ocean sectors (Figure 2b,e). The sea ice in the Southern Ocean acts as a barrier to heat exchange with the atmosphere. Hence, low SIC and SIT denote that the cooling after 2006 is possibly due to enhanced surface heat loss. Cooling and salinity increase in both the surface and intermediate layers of the Weddell Sea sector before 2006 (Figure 8b,e) when considered together with the continuous warming in the bottom layer (Figure 8c), reveal an important feature since it allows for vertical stratification to weaken, thus favoring deep convection. Deep convection would then transfer the low-temperature signal to the bottom layer. The continuous increase of the salinity in the intermediate layer of the Weddell Sea sector from the beginning of the reanalysis is also an important feature that points to either an overestimation of sea ice formation throughout the years or a continuous formation of saline water masses.

Before 2004, the standardized temperatures in UR025.4 show a slight warming trend in the bottom layer of all three sectors. After, an intense warming is present from May to October of 2004, when the bottom layers of both the Indian Ocean and Western Pacific sectors get three times warmer over this six-month period (Figure 9c). Although this rapid warming signal is not noticeable in the Weddell Sea bottom layer, a long-term warming is still present, mainly after 2004. The intermediate layer in the Western Pacific sector shows the same intense warming signal, however from May to December of 2004, while the warming of the intermediate layer is not very noticeable in the Indian Ocean and Weddell Sea sectors (Figure 9b). The surface layer temperatures of all sectors do not show any distinct anomalies (Figure 9a).

The salinity anomalies in UR025.4 show alterations during the AABW formation period (2004) simultaneously with the changes in temperature. Before 2004, the bottom layer salinity appeared to decrease slowly (Figure 9d). In April and May of 2004, an intense decrease in the salinity anomaly is first seen in the surface and intermediate layers of the Indian Ocean sector, and low salinities are also recorded in the intermediate layer of the Western Pacific sector during this period (Figure 9d-e). After May 2004, the salinity anomalies begin to increase sharply until August, and grow by 4 anomaly units in the surface layer and 6 anomaly units in the intermediate layer of the Indian Ocean sector, while the anomalies in the Western Pacific sector grow by 3 anomaly units in the intermediate layer (Figure 9d-e). The bottom layers of both the Western Pacific and Indian Ocean sectors show an increase in salinity between May and September of 2004, which denotes a downward propagation of saline waters into the intermediate layer (Figure 9f). The Weddell Sea sector also has a high salinity signal, but instead of occurring in 2004, the salinity increase is gradual and more noticeable from 2004 until the end of the time series.

The temperature and salinity anomalies in the layers of UR025.4 in 2004 show some important mechanisms that occur along with AABW formation. First, the sharp positive peak in the salinity anomalies in the entire water column of the

Indian Ocean sector (Figure 9d-f) suggest that an intensified brine release occurred from May to October of 2004. This period coincides with Austral Winter; thus, this brine release might be directly connected to enhanced sea ice formation. Although sea ice increase is not seen in the annual averages, that might be due to a short period and location of sea ice formation in the model (explained in section 3.4). Furthermore, the fact that the high salinity signal is also seen in the bottom layer (Figure 9f) shows that the surface buoyancy loss due to brine release was enough to increase the water mass density to neutral bottom water densities, which indicates the importance of brine release in AABW formation in UR025.4. In fact, neutral density contours along Prydz Bay show salinity anomalies increasing and being exported to the bottom layer as SIT anomalies grow (Supplementary 1). It is important to highlight that from January to May of 2004, i.e., before the brine release event, extremely low salinity anomalies are recorded concomitantly in the surface and intermediate layers of the Indian Ocean sector (Figure 9d,e). Since a freshening signal lowers the water mass densities, this signal tends to not propagate downwards from the surface. Hence, we believe that the freshening started in the intermediate layer and propagated to the surface. Lateral decreases in the salinities of the waters isolated from the surface are probably connected to advection and water mass mixing. We believe that those processes may be the drivers of the freshening signal in the intermediate layer.

Finally, the second alteration evident in the UR05.4 anomalies was a warming of the Western Pacific and Indian Ocean sector bottom layers between May and October of 2004 (Figure 9c).These anomalies indicate that the high-salinity water mass that was exported as bottom water also had a higher temperature signal. Moreover, it seems clear that salinity increase at the surface due to brine release is, in fact, the direct mechanism of AABW formation in UR025.4 in the Indian Ocean and Western Pacific sectors, as it compensates for the associated warming also present in those sectors. In the Weddell Sea sector, a salinity increase and warming are also seen, but as a steady long-term growth in the bottom layers after 2004 (Figure 9e,f). This increasing salinity in the bottom layers is likely a result of the intense sea ice formation in the Weddell Sea (Section 3.1).

Finally, in SoSE, the WSBW and WSDW formation occur mostly during the first few months of the time series in 2005, with smaller total formation pulses (WSBW +WSDW) in 2006 and 2007 also (Figure 3- WSDW and WSBW). Therefore, it is not possible to evaluate the conditions before bottom water formation. WSBW and WSDW formation also occurred only in the Weddell Sea, so the analysis is focused on this sector. During the WSDW formation in 2005, the bottom layer of the Weddell Sea sector experiences a warming and a salinity increase from May to June, showing that the relatively high temperature and salinity are characteristic of the WSDW formed in the SoSE (Figure 10c-f). The surface and intermediate layers of the Weddell Sea experience intense cooling (Figure 10a,b), which together with the bottom layer warming could lead to diminished stratification. Different from ECCO2, however, the salinities in the intermediate layer of SoSE diminish (Figure 10e), which can again favor stratification. Thus, we can say that the mean temperature and salinity anomalies by sector do not necessarily point to a broad deep convection event. No salinity increase consistent with the intensified brine release is evident in the layers of the Weddell Sea sector either, so further analysis is necessary to determine the mechanism of AABW formation in the SoSE.

## 3.4 Modeled Bottom Water Formation

The anomalous signals identified by the average SIC and SIT distribution in ECCO2 are mainly connected to the appearance of a large-scale sensible heat polynya the Weddell Sea sector (Figure 11a-c) and the neutral density alterations (Figure 11d-f), as previously pointed out by Azaneu et al. (2014). The polynya begins to open in November 2003 near 20°E, and spreads into the Weddell Sea, Indian Ocean and Ross Sea sectors. That anomalous process is clearly evident by the sea ice concentrations lower than 30% and the lack of accumulation of sea ice (Figure 11g-i). This low sea ice content signal is extreme enough to decrease the monthly mean sea ice concentrations and thicknesses in the Weddell Sea and Indian Ocean sectors, and even the whole Southern Ocean average (Figure 2a,b,e). During the winters without open ocean polynyas, the sea ice cover acts as a thermal barrier that hinders heat exchange between the seawater and the cold winter atmosphere. Therefore, when the ECCO2 polynya opens at the end of 2003, the heat exchange through the water surface increases. With the intensive cooling over the polynya during the following winter, the heat loss to the atmosphere causes the water buoyancy to decrease, which ultimately creates WSDW from 2004 to 2008 (Figure 3 – WSDW), as evidenced by the increase in WSDW volume explained in section 3.2. The temperature and salinity decreases described in section 3.3 in the bottom layer of ECCO2 after 2006 are related to the polynya appearance and AABW formation, since the bottom layer of the Weddell Sea in this reanalysis essentially contains AABW. Hence, since cooling and freshening is present in bottom layer of Weddell Sea sector in ECCO2, we suggest that AABW varieties formed under the Weddell polynya retain the distinct low salinity and low temperature signals due to heat loss at the surface. These water masses are then exported to the intermediate and bottom layers of the Weddell Sea, Indian Ocean and Western Pacific sectors.

Furthermore, maps of the neutral densities in the intermediate layers of ECCO2 show continuously increasing formation of WSDW offshore due to cooling at the prime meridian during the following winters (Figure 11d-f). After 2008, with the expansion of the polynya, the heat lost through the surface becomes even stronger, and both WDW and WSDW cool even further to form WSBW (Figure 11f). That expansion leads to 70% of the Weddell Sea sector filled with WSBW volume by the end of 2013 (Figure 3 – WSBW). Due to limited data sampling, real ocean monthly estimates of WSBW variability are not currently possible. However, some efforts have been made by previous studies to account for the average contribution of WSBW to the Weddell Sea sector. Pardo et al., (2012) used extended Optimum Multiparameter Analysis (eOMP) to quantify the volumes of the Southern Ocean water masses and found that the longitudinal limits of our Weddell Sea sector was filled with approximately 26±0.2% of WSBW, a percentage substantially lower than the 70% of WSBW estimated by ECCO2 in 2013. This previous article uses 45°S as the northern limit for the volume calculations, while our calculation uses 60°S, which accounts for some of the difference in the volume values.

It is important to highlight that the timing of the signals in SIC, SIT, temperature, salinity and neutral density are different. First, even though the polynya appearing in ECCO2 opens in November 2003, the signal of decreasing SIC and SIT appear only from 2004 onwards. That is because the sea ice data used in this study are annual averages, and since the polynya only opened at the end of 2003, its signal was not enough to diminish the SIC and SIT annual averages. Also, even

though the polynya was already established in Weddell Sea in 2004, the freshening and cooling signals in the bottom layer of ECCO2 are noticeable only after 2006 (Figure 8c and 8f). Again, that is because the monthly average temperatures and salinities were calculated, for each cell, as a mean weighted by the volume of the cell. Hence, the signals in temperature and salinity only appear in the bottom layer after the new volume of the bottom water has been replaced. Finally, it is also important to note that, even though the Polynya had opened in November 2003, the bottom water production (WSDW and WSBW) signal appeared in the intermediate layer in 2007 onwards (Figure 11e).

The decrease of sea ice that contributed to the polynya opening in ECCO2 could have been caused by several factors, such as changes in the balance of the heat in the atmosphere and the heat supplied by the ocean under the mixed layer (Close and Goosse, 2013), and the mixing of warmer and saltier intermediate waters with surface waters (Morales-Maqueda et al., 2004). In ECCO2, the process that opened the polynya was due to water mass alterations and differential heat delivery to the surface. Starting from 1992, ECCO2 experiences an increase in the volume of WDW in the Weddell Sea and a decrease in AASW (Figure 3- AASW and WDW), which causes a WDW isopycnal uplift to shallower depths until 2000 when WDW reaches the surface. The increase in salinity anomalies in the intermediate layer of the Weddell Sea sector is also an expression of the WDW volume increase in this sector since this water mass has distinctly higher salinities than the waters above it (Figure 8e). After reaching the surface, WDW begins to exchange heat with the local sea ice and atmosphere, the water cools and is slowly converted into WSDW during the first four years. By the winter of 2004, the heat transported by the WDW to the surface is enough to melt sea ice and open the large polynya near the eastern limb of the Weddell Gyre, approximately 20°E (Figure 11a). After the polynya opens, a more intense conversion of WDW to WSDW occurs until 2008, when the latter starts to transform into WSBW (Figure 3 – WSDW and WSBW). This four-year delay between the year when WDW reaches the surface and the opening of the polynya is very close to the five-year estimate of the residence time of WDW in the Weddell Gyre before it is converted to denser local water mass varieties (Fahrbach et al., 2011). This mechanism that explains the polynya opening is the same one believed to have triggered the 1970s Weddell Polynya (e.g., Killworth, 1983; Cheon et al., 2015).

The SoSE reanalysis also shows the presence of an open ocean polynya in the first year of simulation in the Weddell Sea sector (Figure 12a), which persists from May to November 2005. From January to May, before the polynya opens, the presence of WDW at 10 m at approximately 70°W is noticeable in the SoSE neutral density transects. The upper limit of WSDW is located at approximately 1500 m depth, and the WSBW surface is at 4000 m depth (Figure 12b). The neutral density transect in August 2005 after the polynya opened shows a shift of the WSDW boundary towards the surface at approximately 70°W (Figure 12c). This shift shows the conversion of WDW to WSDW over the polynya, as seen in the previous section. WDW has a higher temperature and salt content than the local overlying AASW. The heat loss in the polynya during the winter reduces the buoyancy of WDW, which results in the formation of WSDW from May to November. By December, the polynya closes, and the WSDW formation slows down. WSDW formation also occurs during the following two winters, but with volumes less than half of the 6% production in 2005.

The timing of the events in SoSE are more tied together, but that is because as soon as the polynya opens, WSDW is formed and transported to the bottom layer (Figure 12c), thus having minimum lag between the ice-free area opening and the WSDW formation. Hence, we can see prior and during the polynya opening a clear warming of the bottom layer and cooling of intermediate layer, which weakened vertical stratification and allowed WSDW to be transported downwards.

The trigger of the polynya in SoSE is similar to that in ECCO2 and was the heat delivery to the surface level by the WDW. The mean surface temperature calculated under the polynya (August 2005), is a degree higher than the calculated for August 2008 when there are no ice-free areas, and crosses the freezing point of seawater. Different from ECCO2, WDW in SoSE is present at the surface before the winter (Figure 10b). With the advancement of the sea ice in the winter of 2005, the WDW enduring high heat content at the surface delays sea ice formation until December, and as a result, an elongated
polynya occurs in the Weddell Sea. Therefore, the mean sea ice thickness in the Weddell Sea sector is the lowest of the entire SoSE time series in 2005 (Figure 2h). The average temperature and salinity anomalies in SoSE do not show a strong indication of weakened stratification, which is possibly because the volume of WSDW created under the polynya is considerably lower than the whole volume of the Weddell Sea sector. In any case, the lower stratification in SoSE might be one of the causes that constrained the polynya to a small area and period.

It seems that WDW uprising is the main mechanism responsible for melting the sea ice, and creating the Weddell polynya in both ECCO2 and SoSE. Although out of the scope of this study, some processes can cause isopycnal uplift in Weddell Sea, creating the open ocean polynya. An experiment with global ocean-sea ice model performed by Hirabara et al., (2012) have suggested that a saline surface layer and persistent cyclonic wind stress are necessary to lower vertical stratification and allow WDW uplift over the Maud Rise. In a recent attempt to reproduce the Weddell Polynya, Cheon et al.,
(2015) has found that the establishment of strong negative wind stress curl in Weddell Sea accelerates the Weddell gyre, causing WDW to upwell in the center of the gyre and melting sea ice. Furthermore, a simulation with the Kiel Climate Model (KCM) have shown that warm waters built-up in Weddell Sea deep layer during non-convective periods, and after decades the heat buffered interact with sea ice opening the Weddell Polynya (Martin et al., 2013).

Some processes other than the ones analyzed in this study may have influenced the polynya opening in the
abovementioned reanalysis. Parkinson (1983) modeled the 1976 observed Weddell Polynya and noticed that the wind patterns seemed to control whether the polynya would open or not in the simulation. Additionally, some simulations have shown that the interactions of currents and eddies with the Maud Rise can alter the oceanic transport throughout the whole Weddell Sea water column, which allows heat exchange with the sea ice and opens the Weddell Polynya (Holland, 2001a, 2001b). In a more recent study, Gordon et al. (2007) noticed that during long periods of the negative Southern Annular Mode
Index (Limpasuvan and Hartmann, 1999), the Weddell Sea had enhanced sea ice formation and brine release, which destabilized the water column and allowed the Weddell Polynya to occur. Lavergne et al. (2014) also stated that the reduced occurrence of open ocean deep convection in the Southern Ocean after 1950 was due to strong ice melt and enhanced Southern Ocean stratification, which indicated that strong sea ice seasonality is an important trigger for polynya opening. In that matter, a recent study found that GISS-E2-R and GFDL-ESM22 ocean models from CIMP5 attribute the frequent deep

convection events to stronger sea ice seasonality. This stronger seasonality enhances winter brine release and homogenizes the Weddell Sea water column, which allows deep convection to transfer heat to the surface (Heuzé et al., 2015). According to Azaneu et al. (2014), ECCO2 also exhibits strong sea ice seasonality and that likely plays a role in the opening of the ECCO2 polynya. Despite none of those specific atmospheric and sea ice patterns being analyzed here, their manifestation

allows WDW to rise to the surface leading to deep convection and thus acting as triggers to polynya establishment. Moreover, several studies show the same pattern of WDW raising to the surface, exchanging heat and opening the Weddell polynya in simulations (e.g., Hirabara et al., 2012; Martin et al., 2013; Cheon et al., 2015). It is also worth mentioning that both the ECCO2 and SoSE reanalysis products use the same MITgcm ocean model. Hence, the polynya opening in the Weddell Sea might be an expression of the same features of that circulation model. Finally, because the polynya in SoSE

occurs at the beginning of the reanalysis output, we cannot assure its opening is a result of an initial adjustment process, even though a one year spin-up procedure is conducted in the prior year (2004) to bring the SoSE to its equilibrium conditions (M. Mazloff, personal communication).

Although two of the reanalysis products used in this study yielded open ocean polynya events that formed AABW, the UR025.4 formed AABW in a completely unique way. In this model, the first adjustment observed is a substantial rise in

UCDW volume in the Indian Ocean sector in 2004, as described in the previous section (Figure 4 – UCDW). As seen in Figure 13, this water entered the coastal regions of the Indian Ocean sector from the surface up to the 250-m level and interacted with the ice edge in that region, which caused some of the ice to melt and initially resulted in a freshening signal at the surface and the intermediate layers (Figure 9d,e). That colder and fresher water resulting from the melting was transported westward through the Antarctic Coastal Current (ACoC). In the Southern Ocean, the CDW-derived waters that

enter Prydz Bay interact with ice shelves and sea ice and form Dense Shelf Water (DSW). This process in turns leads to the export of DSW down the slope of Prydz Bay, which mixes with denser CDW layers and forms the regional AABW variety (Williams et al., 2016). The water mass pulses seen in the UR025.4 Indian Ocean sector suggest a process similar to this real process. Both in the reanalysis and in reality, UCDW is warmer and fresher than the regional LCDW that circulates along the coast. Hence, while the UCDW coastal entrainment in UR025.4 (Figure 13c,e,g) circulates along the sea ice edge, it

contributes to the melting of sea ice, lowers the surface salinity even further (Figure 9d-e), and is transported westward through the ACoC. Those waters that result from the mixture of UCDW and melting sea ice have higher freezing temperatures. Thus, as the waters circulate along the ACoC in UR025.4, they enhance sea ice formation and release brine further along the way, i.e., over LCDW waters. That process concentrates the salt over the LCDW, increasing its density and forming DSW by continental shelf convection (Figure 13). The salinization by sea ice formation in this process is evident in

the surface and intermediate layers of the Indian Ocean sector, and in the intermediate layer of the Western Pacific sector from May until August of 2004 (Figure 9d-e).

After formation, the DSW in UR025.4 flows down the slope, enhancing LCDW salinity and creating a very saline variety of AABW. The high salinity signal seen in the bottom layer of both the Western Pacific and Indian Ocean also indicates the exportation of this saline DSW to the bottom layers and the formation of AABW (Figure 9f). This conversion is

responsible for the decrease in the UDCW and LCDW volumes in the Indian Ocean sector water masses time series. Hence UR025.4 rather accurately represents both the warm water entrance into Prydz Bay and the density increase along the circulation present in the real world. The small excess of UCDW that does not interact near Prydz Bay in UR025.4 is further carried through the ACoC to the southernmost Weddell Gyre border. The excess UCDW combines with the underlying high salinity LCDW and forms part of the AABW. This advection of AABW from the Indian Ocean sector is a process that has previously been found to occur in the real ocean (Jullion et al., 2014).

Thoroughly inspecting monthly maps of salinity and temperature anomalies (not shown) revealed that UCDW originating from the Indian Ocean entering the Weddell Sea in UR025.4 is colder and with lower salinity than the local WDW in the Weddell Sea, especially due to the melting episode that occurred in Prydz Bay in early 2004. As a result, a slight freshening signal is carried through the ACoC in the intermediate layer of the Weddell Sea sector and reaches the Antarctic Peninsula at the end of 2005 (Figure 9e). Similar to the process occurring in the Indian Ocean sector, the waters with lower salinities facilitate sea ice formation and increase sea ice content along the way. In fact, after 2007, the SIT on the eastern Antarctic Peninsula increases and reaches values greater than three meters in 2010 (Figure 14i). This ice-thickening event is also marked in the annual average SIT of the Southern Ocean (Figure 2h). Before the thickening event, WSDW is present at approximately 700 m only in a small region east of the Antarctic Peninsula, while WDW takes up the majority of the Weddell Sea (Figure 14d). After the thickening event in 2007, the rejected brine follows the enhanced SIT and produces an HSSW pulse in the Weddell Sea, which mixes with the local WDW and WSDW and creates WSDW and WSBW (Figure 14e). Especially in 2010, the brine release becomes so intense that it creates WSBW varieties from the eastern Antarctic Peninsula to the prime meridian (Figure 14f). Furthermore, the increased sea ice thickness in the Weddell Sea could also be due to the advection of sea ice from the Indian Ocean and the Western Pacific sectors, as will be explained further. Although the process that occurs in the Weddell Sea sector is different from the one in the Indian Ocean sector, the slow and steady salinity increase from 2004 until 2010 in the bottom layers of the Weddell Sea sector is also due to the long-term advection of saline AABW and LCDW from the Indian Ocean sector (Figure 9f).

Fresh UCDW from ice melting in the Indian Ocean sector actually entered the Western Pacific sector from the surface down to the 250-m level, although the volume was not enough to show up in the total UCDW volume series (Figure 13c,e,g). An eastward density gradient directed along the coast is created by the density difference between the neutral densities of UCDW and LCDW. In response to that gradient, a strong baroclinic current opposite the gradient, i.e., a western current, appears at the surface in UR025.4 in Vincennes Bay (Figure 13d). In April, the area surrounding Vincennes Bay (66.5°S, 109.5°E) is covered by sea ice in UR025.4; thus, a reasonable cause of the current intensification is the temperature/salinity gradient between UCDW and LCDW. The zonal speeds during the speed-up period reach values up to 0.09 m s$^{-1}$, which is the highest speed in Vincennes Bay in all reanalysis periods (Figure 13f-g). Moreover, the ACoC strengthens after the UCDW inflow, and the northeast directed current is formed by May (Figure 13f, also Supplementary 3). Sea ice speeds also increase as the current speed rises. Relatively fresh UCDW from sea ice melt was then transported by both the strong ACoC and the offshore-directed buoyancy current, losing heat, enhancing freezing and releasing brine over

both CDW varieties, which creates DSW (Figure 13a,c,e). After June, (Figure 13g-h), the intense DSW formation seems to recirculate next to Vincennes Bay and is exported northwest also due to a strong southeast density gradient. The high density of DSW then leads to convection next to the continental shelf, with the water flowing down the slope, mixing even more with LCDW, and thus forming AABW. Hence, the higher inflow of UCDW in both the Indian Ocean and Western Pacific sectors triggers a balance between melting and ice formation that favors LCDW salinity increases and speeds up the ACoC.

Although the UR025.4 reanalysis creates AABW in the Indian Ocean sector through a mechanism similar to the one proposed by Kitade et al. (2014) and Williams et al. (2016), it still has significant differences. First, the real AABW formation in the Indian Ocean sector occurs after the modified Circumpolar Deep Water (mCDW) circulates deeper under the ice shelves surrounding Prydz and Vincennes Bays, mixing with the DSW created in the coastal polynyas and increasing its salt content as well as lowering its temperatures (Williams et al., 2016). The mCDW has neutral densities between 28.0 kg m$^{-3}$ and 28.27 kg m$^{-3}$; hence, relatively small salt input or cooling could easily diminish the buoyancy of mCDW, thus forming AABW. In contrast, the UR025.4 water mass in Prydz Bay (UCDW) is relatively warmer, less saline and lighter than mCDW, which causes a balance between immediate melting under the sea ice and freezing as the water circulates along the coast. This balance releases brine over LCDW and creates very saline LCDW and AABW varieties. In addition, the AABW formation in UR025.4 occurs mainly by salinization, and as a result, the final AABW has substantially higher salinity than expected. Additionally, the increase in density caused by this salinization combined with the relatively low density of UCDW causes a baroclinic current that enhances the speed of the ACoC and creates a current directed offshore that exports DSW downslope from the shelf.

With respect to timing, signals in sea ice variables and ocean variables in UR025.4 had minimum lag. That is because the AABW is formed here rather abruptly. SIT annual averages only reached its peak in Weddell Sea in 2009, even though the AABW formation in this sector occurred in the winter of 2008. However, the sea ice increase in the annual average is noticeable from 2006 until 2009, showing that the major sea ice production happened during this period, which is in agreement with the AABW formation in 2008 at Weddell Sea Sector by brine release. Regarding the salinity anomaly signals in the Indian Ocean Sector, it is possible to see that the minimum salinities happened almost simultaneously in the intermediate and surface level (Figure 9 d and e). That is because the UCDW entrainment responsible for this fresh signal happened simultaneously in the intermediate and surface layer. Moreover, the following high salinity signal in all three layers are simultaneous (Figure 9 d-f), as the newly formed bottom water is swiftly injected to the bottom layer (Supplementary 2). Finally, pulses of WSDW production were described in Weddell Sea sector from 2005 until 2008, however their magnitude were too small to print signatures in temperature and salinity mean values.

Finally, in all three reanalysis products investigated in this study, the AABW formation occurred due to entrainment warm CDW-derived waters and interaction with sea ice. Why then the mechanism of AABW formation in UR025.4 is different from the other two reanalysis (ECCO2 and SoSE) ? One of the possible explanations might be that the advection of CDW-derived waters in UR025.4 originates from the east in the Weddell Sea. There is a region with consistently low sea ice concentrations and thicknesses near the center of the Weddell Sea, which is due to the natural isopycnal uplift inside the

Weddell Gyre (de Steur et al., 2007). Hence, the warm CDW waters that flow west along the isopycnals tend to rise when they reach the central Weddell Gyre, while they stay roughly at the same depths when they flow east towards the Indian Ocean sector and only upwell along the coast due to coastal divergence. Thus, the warm water in the deep Weddell Sea layers is expected to exchange heat with the sea ice in the central Weddell Gyre, which can likely lead to the establishment

of a polynya. In fact, Timmermann and Beckmann (2004) attempted to accurately reproduce that so-called *warm water halo* and found an enhanced vertical heat exchange with sea ice, which resulted in the opening of an oceanic polynya in the Weddell Sea. In addition, long-term cooling of the intermediate layers and the warming in the bottom layers of ECCO2 might have played a role in polynya establishment. Those trends decrease Southern Ocean vertical stratification and allow heat to be transported upwards and deep convection to happen. Azaneu et al. (2014) discussed the possible triggers of the

ECCO2 polynya and suggested that the long-term warming of the bottom waters was one of the main factors that contributed to the polynya establishment and subsequent expansion. Long term warming of bottom waters have been pointed as a trend also in other non-assimilatory models, such as the Kiel Climate Model (Martin et al., 2013) and Climate models CM2.5 and CM2.6 (Dufour et al., 2017), and in both studies the heat buffered has opened a polynya in Weddell Sea. In addition, both ECCO2 and SoSE use the same ocean model and similar modeling frameworks, so we cannot rule out the appearance of the

polynya in both models as an expression of similar model features of the reanalysis products.

## 4 Summary and Conclusions

Deep convection in open ocean polynyas, together with sea ice and ice shelf representation, have been identified as frequent causes of spurious AABW formation and a source of erroneous ocean representation in non-assimilatory OGCMs (e.g., Kerr et al., 2009b; Renner et al., 2009; Meccia et al., 2013; Stössel et al., 2002). Spurious open ocean deep convection creates

AABW through a rather abrupt mechanism and enhances bottom layer ventilation and heat/carbon uptake, which inserts errors in climate estimates (e.g., Heuzé et al., 2013). The propagation of spurious open ocean deep convection in the Southern Ocean simulations can go even further and cause the warming of the abyssal layer, the cooling of surface waters and atmospheric warming (Latif et al. 2013; Pedro et al. 2016). The mechanisms of bottom water formation in some of the reanalysis investigated here agree with those ideas: two of the three reanalysis products formed AABW by deep convection

through the appearance of open ocean polynyas. In both ECCO2 and SoSE, the presence of WDW at the surface hinders sea ice formation during the winter and triggers a deep convection event, which creates AABW by cooling. Especially in ECCO2, the mechanism of AABW formation resulted in erroneous representations of the Southern Ocean, such as high AABW volumes and lower sea ice concentrations and thicknesses, reinforcing that open ocean deep convection inserts errors in the simulation (e.g., Azaneu et al., 2013; this study). Although the appearance of open ocean polynyas in models without

assimilations is a well-known modeling issue, the opening of the Weddell Polynya in the reanalysis products is a new documented feature. Since the reanalysis products are meant to be precise representations of the ocean state due to the assimilation of observations, the appearance of spurious AABW formation by open ocean deep convection shows that the

assimilation of real ocean sea ice variables has not been enough to hinder the appearance of open ocean polynyas. Furthermore, weak stratification that enhanced WDW heat release to the surface seemed to be one of the main triggers of the Weddell Sea Polynya opening in the ECCO2 and SOSE reanalysis products. In this matter, the low spin-up period of 1 year in SoSE and lack of spin up in ECCO2 could have allowed instability in water column in the reanalysis, hence weakening

stratification. However, the WDW increase reported here is consistent with the observed results reported by Kerr et al. (2017 - under review), who found a consistent increase of the WDW contribution to the total mixture of deep and bottom waters in the Weddell Sea from 1984 to 2014, despite the high degree of interannual variability. However, since no real open ocean polynya has been reported since the 1970s (Gordon 1978), a critical analysis of the model mechanisms of heat exchange between the surface waters and sea ice is required in the future to efficiently understand the role of WDW in open ocean

polynya establishment. In addition, since bottom layer warming and intermediate layer cooling are the possible mechanisms that diminished stratification in ECCO2, further evaluation of the causes of those trends is needed to understand the primary factors leading to the weak ocean surface stratification.

On the other hand, UR025.4 created AABW in a different way than that simulated by the ECCO2 and SoSE reanalysis. The highest AABW formation observed in the UR025.4 product occurred mainly east of the Weddell Sea sector. Those findings

agree with the theory proposed by Meredith et al., (2000) that AABW present in the Weddell Sea receives considerable input of AABW varieties from the east, such as the recently observed evidence in the Southern Ocean (e.g., Jullion et al., 2014; Kerr et al., 2017). The entrainment of UCDW in the Indian Ocean sector created an equilibrium between melting and freezing that favored LCDW salinization and led to coastal deep convection. The salinization created DSW that mixed with LCDW and formed AABW while flowing downslope. This mechanism of bottom water formation is dynamically similar to

the real ocean formation of AABW in this region since it depicts both the spilling of DSW off the continental shelf and the exportation to the open ocean (Kitade et al., 2014; Williams et al., 2016). Although this mechanism is closer to the real bottom water formation in the eastern Southern Ocean, the sea ice balance is created by anomalous dynamic processes. First, the density gradient seems to enhance ACoC circulation by creating a buoyancy current. The enhanced transport by the buoyancy current accumulates sea ice around the Antarctic Peninsula and creates unrealistically high sea ice thicknesses,

which in turn induces the spurious salinization of the deep and bottom layers (Dotto et al., 2014). Dense waters with low salinity and temperature are advected from the Indian Ocean sector to the Weddell Sea, changing the properties along the way and interacting with local water masses to form a very saline AABW variety. Nevertheless, the somewhat accurate dynamical representation of the AABW formation in UR025.4 is an advancement in the Southern Ocean representation in OGCMs and ocean reanalysis products since it shows that advanced ocean reanalysis frameworks are getting closer to

efficiently representing AABW formation (e.g., Kerr et al., 2009b, 2012b; Azaneu et al., 2014, Dotto et al., 2014). Furthermore, the substantial differences of AABW formation in the Indian Ocean sector between the UR025.4 simulation and real processes show that improvements in the simulations of sea ice dynamics and ocean-sea ice heat exchange are still necessary to best represent the lower limb of the AMOC.

## 5 Acknowledgements

This study is a contribution to the activities of the Brazilian High Latitudes Oceanography Group (GOAL) and the Brazilian National Institute of Science and Technology of Cryosphere (INCT-CRIOSFERA; 573720/2008-8, 465680/2014-3). The GOAL has been funded by the Brazilian Antarctic Program (PROANTAR) through the Brazilian Ministry of the

Environment (MMA), the Brazilian Ministry of Science, Technology, Innovation and Communication (MCTIC), and the Council for Research and Scientific Development of Brazil (CNPq; 550370/2002-1; 520189/2006-0; 556848/2009-8; 405869/2013-4). W.A. acknowledges the financial support from the CAPES Foundation. M.M.M and R.K acknowledge CNPq grants n° 306896/2015-0 and 302604/2015-4, respectively. We also thank the NSIDC and the research groups responsible for developing the ECCO2, SoSE and UR025.4 and for making their datasets available online. We would finally

like to thank C. Heuzé and the anonymous referee for the valuable suggestions that improved the manuscript.

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

**Table 1: Neutral density surfaces limiting each water mass considered in the study. The asterisk (*) denotes the water mass definitions only applicable in the Weddell Sea sector.**

| Water Mass Acronym | Neutral Density Range (kg/m³) |
|---|---|
| **Surface Waters** | $\gamma^n < 27.7$ |
| **UCDW** | $27.7 \leq \gamma^n < 28$ |
| **LCDW** | $28 \leq \gamma^n < 28.27$ |
| **AABW** | $\gamma^n \geq 28.27$ |
| **AASW*** | $\gamma^n < 28.1$ |
| **WDW*** | $28.1 \leq \gamma^n < 28.27$ |
| **WSDW*** | $28.27 \leq \gamma^n < 28.4$ |
| **WSBW*** | $\gamma^n \geq 28.4$ |

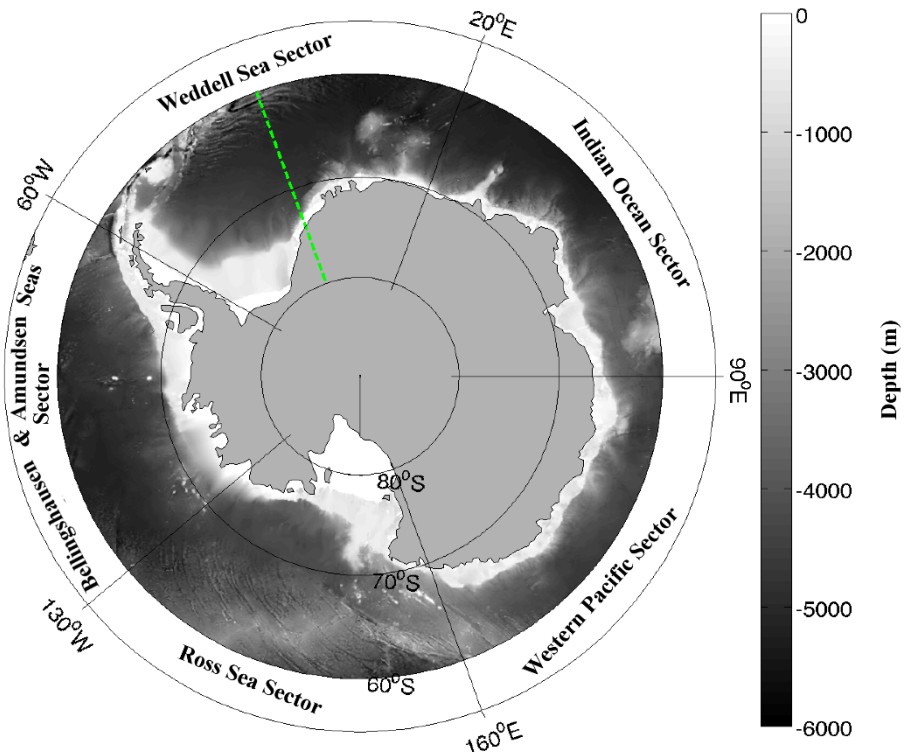

**Figure 1: Sectorial division and bathymetric map of the Southern Ocean. The green line represents the section traced in the SoSE reanalysis. The bathymetric values were retrieved from the ETOPO2 V2 database.**

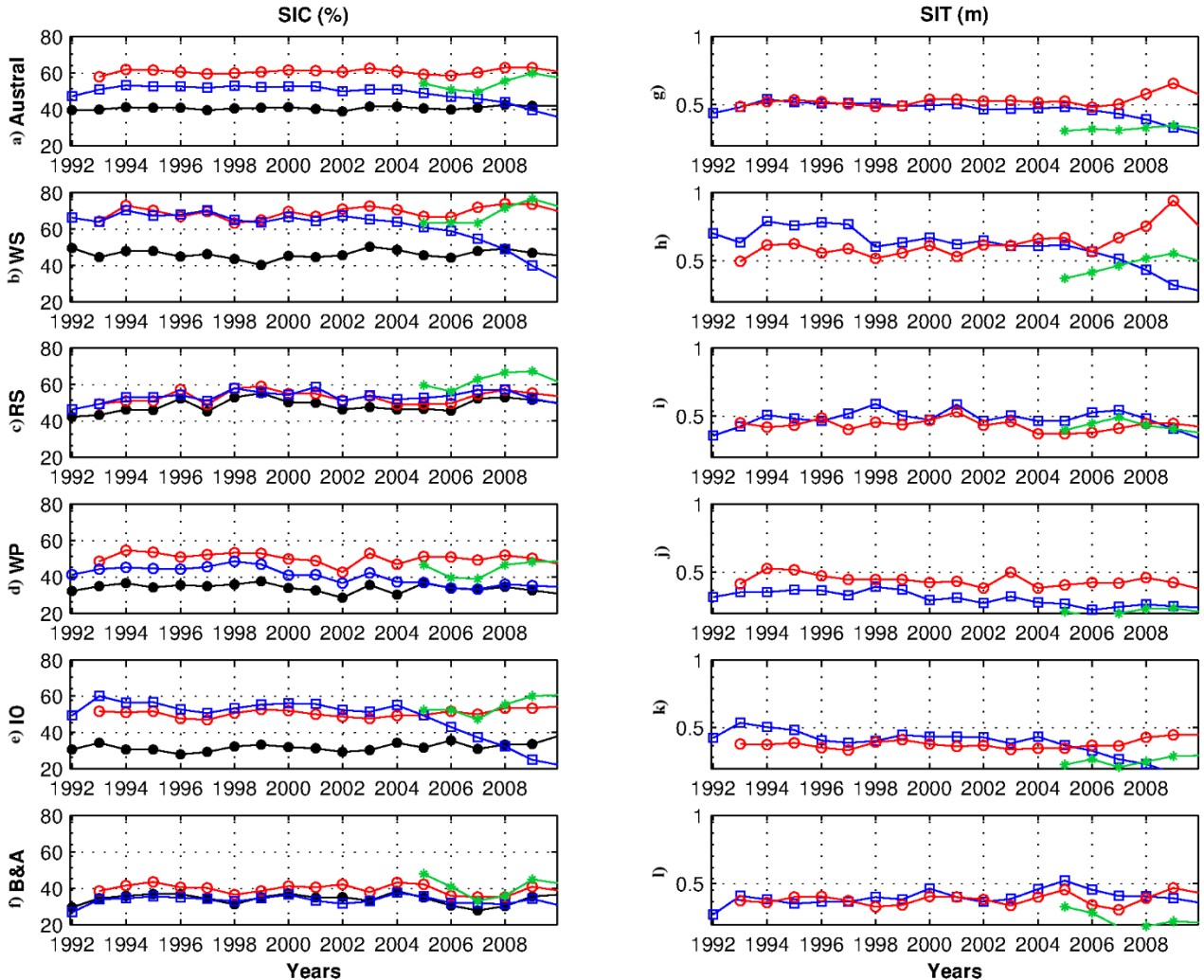

**Figure 2:** Annual mean Sea Ice Concentration (left) and Thickness (right) of each sector of the Southern Ocean. The red(O), green(⋆) and blue (□)lines are the annual mean time seriesfromUR025.4, SoSE and ECCO2,respectively, and the circles, stars and squares are the respective annual values. The black(●) line with the filled black circles represents the mean sea ice concentration from the Goddard Satellite observations. The sectors are labeled WS (Weddell Sea), RS (Ross Sea), WP(Western Pacific), IO(Indian Ocean) and B&A (Bellingshausen and Amundsen Seas).

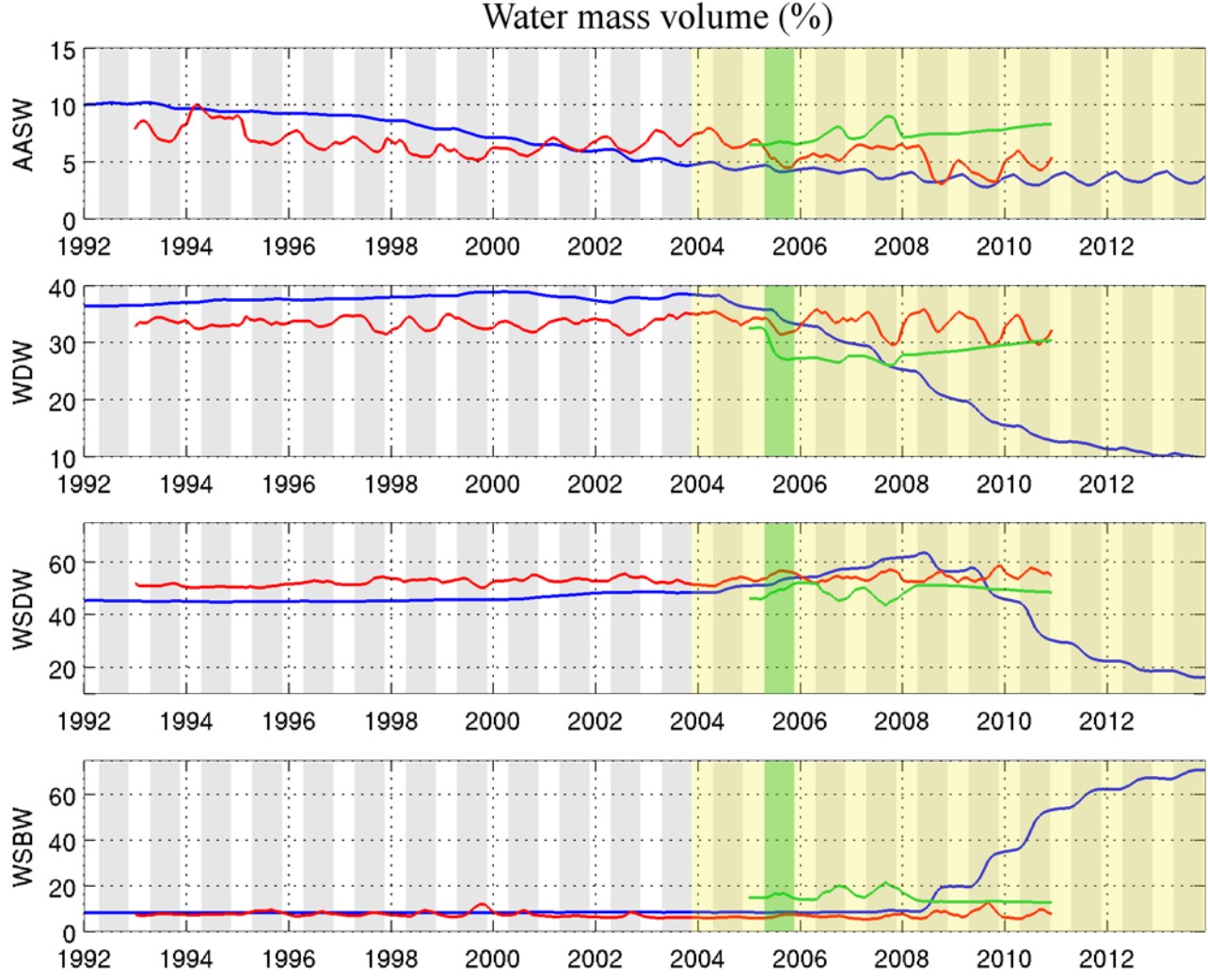

**Figure 3: Percentage of the Weddell Sea sector occupied by the water masses, in volume. From top to bottom, the charts show the volumes for Antarctic Surface Water (AASW), Warm Deep Water (WDW), Weddell Sea Deep Water (WSDW) and Weddell Sea Bottom Water (WSBW). The red, blue and green lines are for the UR025.4, ECCO2 and SoSE reanalyses, respectively. Green and yellow shading highlight the period in which the polynyas are open in SoSE and ECCO2 respectively.**

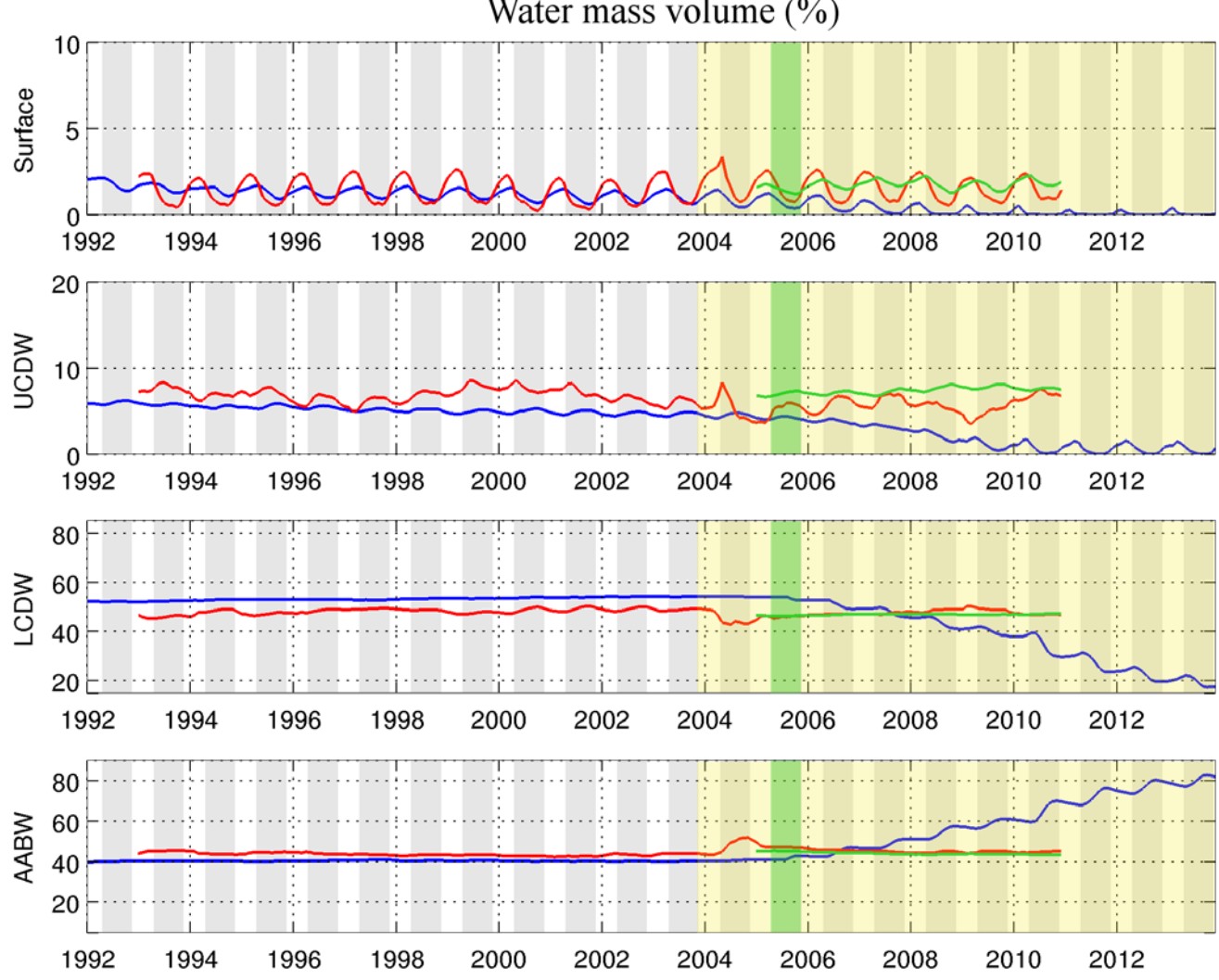

**Figure 4: Percentage of the Indian Ocean sector occupied by the water mass, in volume. From top to bottom, the charts show the volumes for Unclassified Surface Waters, Upper Circumpolar Deep Water (UCDW), Lower Circumpolar Deep Water (LCDW) and Antarctic Bottom Water (AABW). The red, blue and green lines are for the UR025.4,ECCO2 and SoSE reanalyses, respectively. Green and yellow shading highlight the period in which the polynyas are open in SoSE and ECCO2 respectively.**

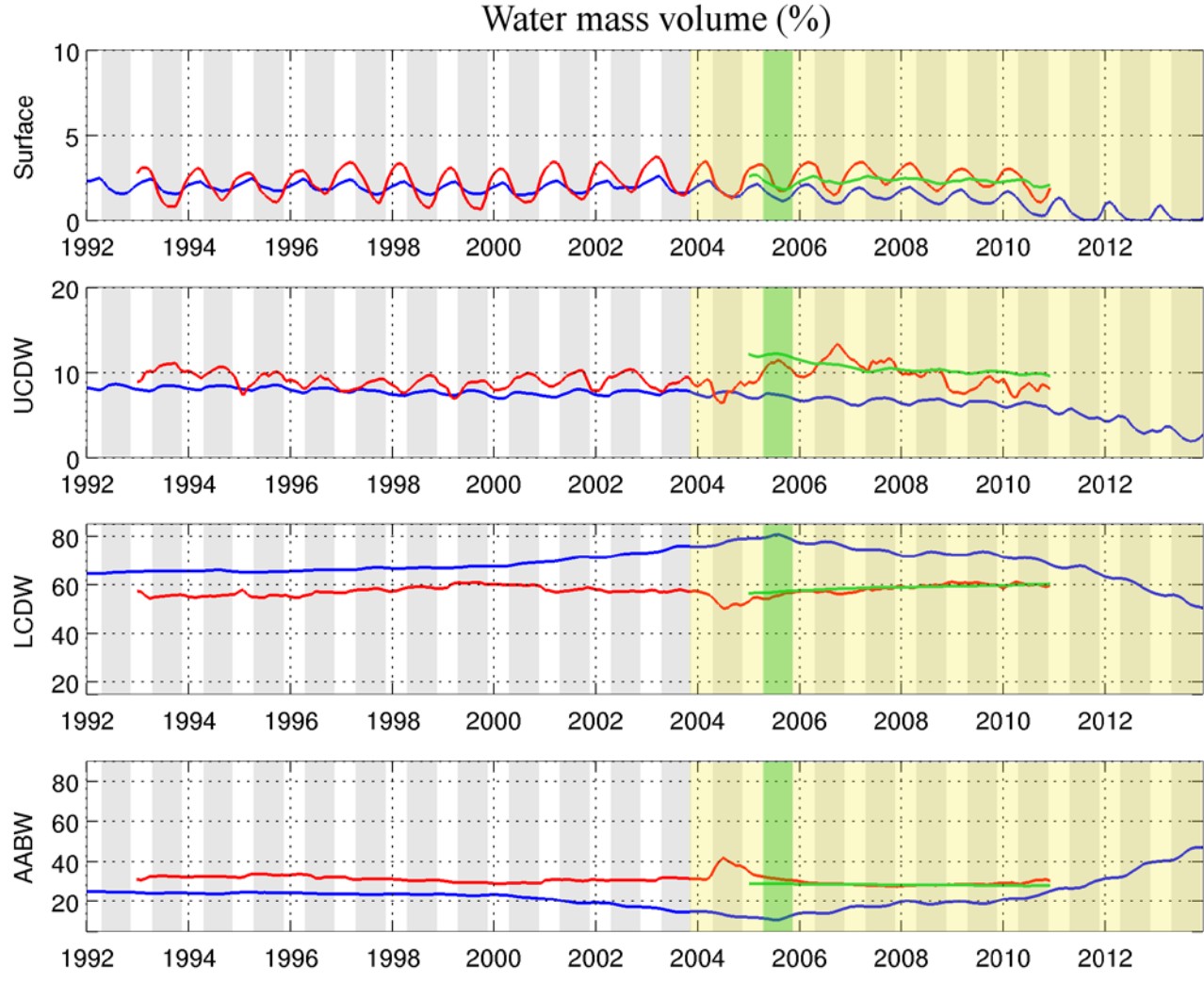

**Figure 5:Same as Figure 4, but for the Western Pacific sector.**

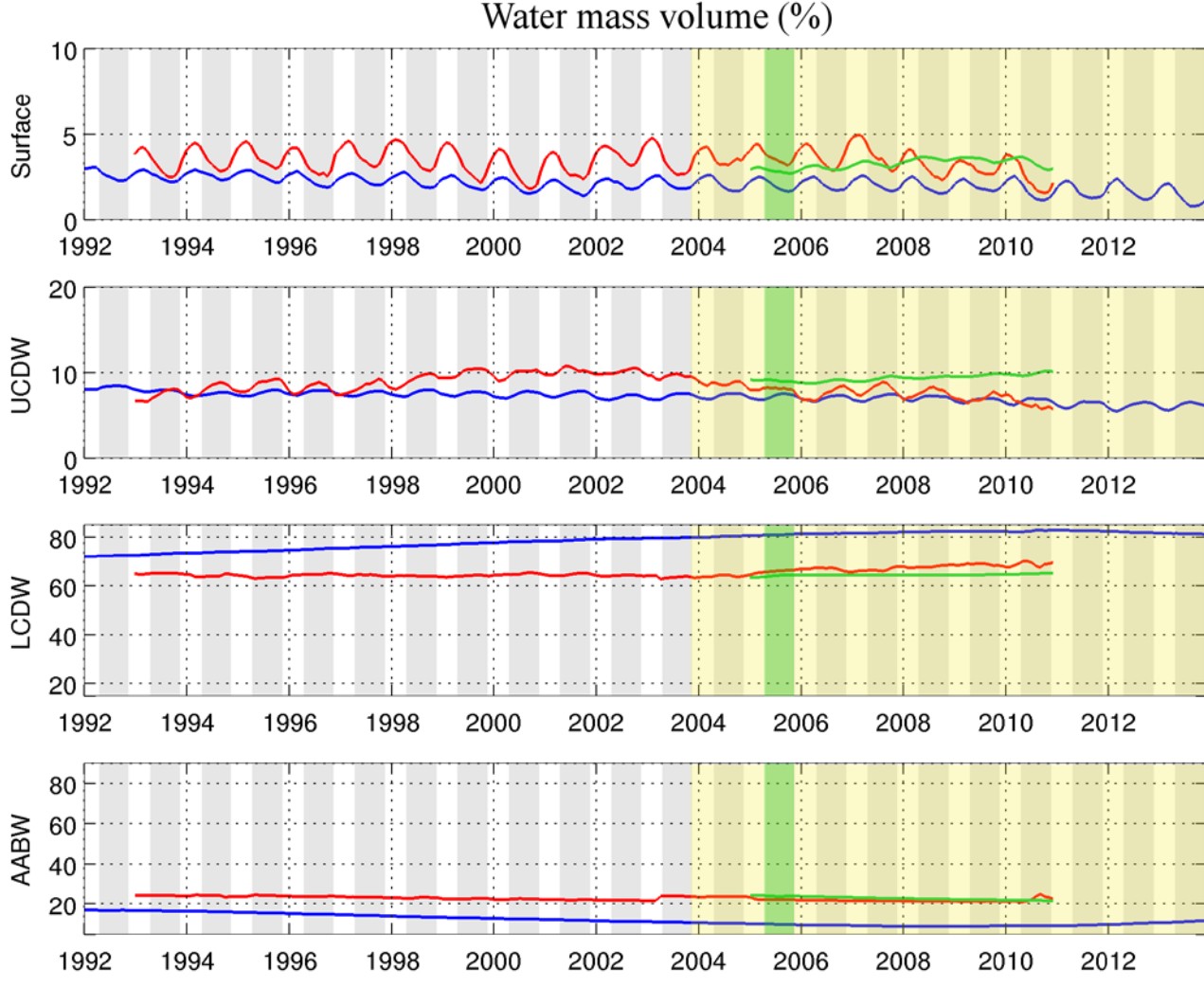

**Figure 6:Same as Figure 4, but for the Ross Sea sector.**

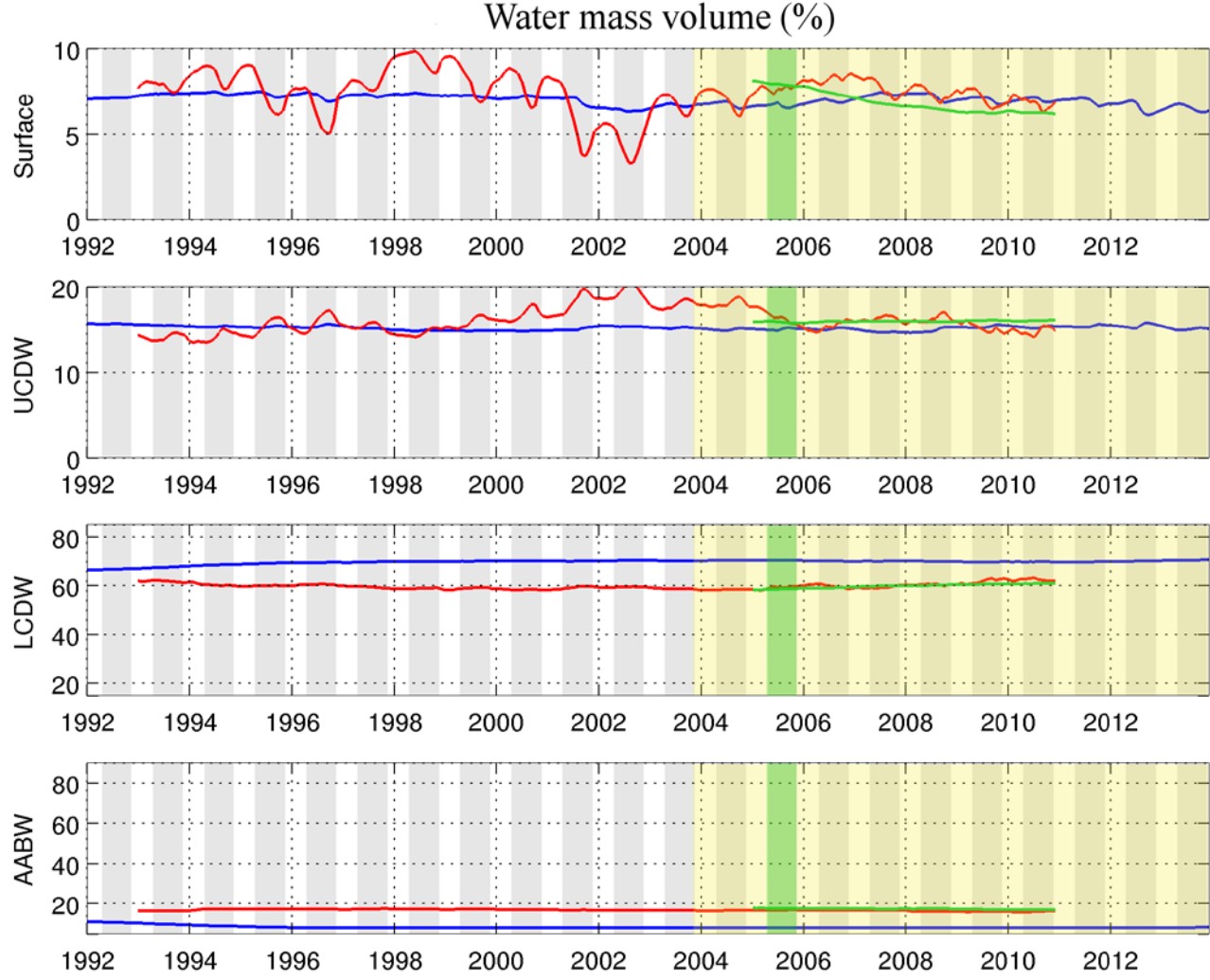

**Figure 7:Same as Figure 4, but for the Bellingshausen and Amundsen sector.**

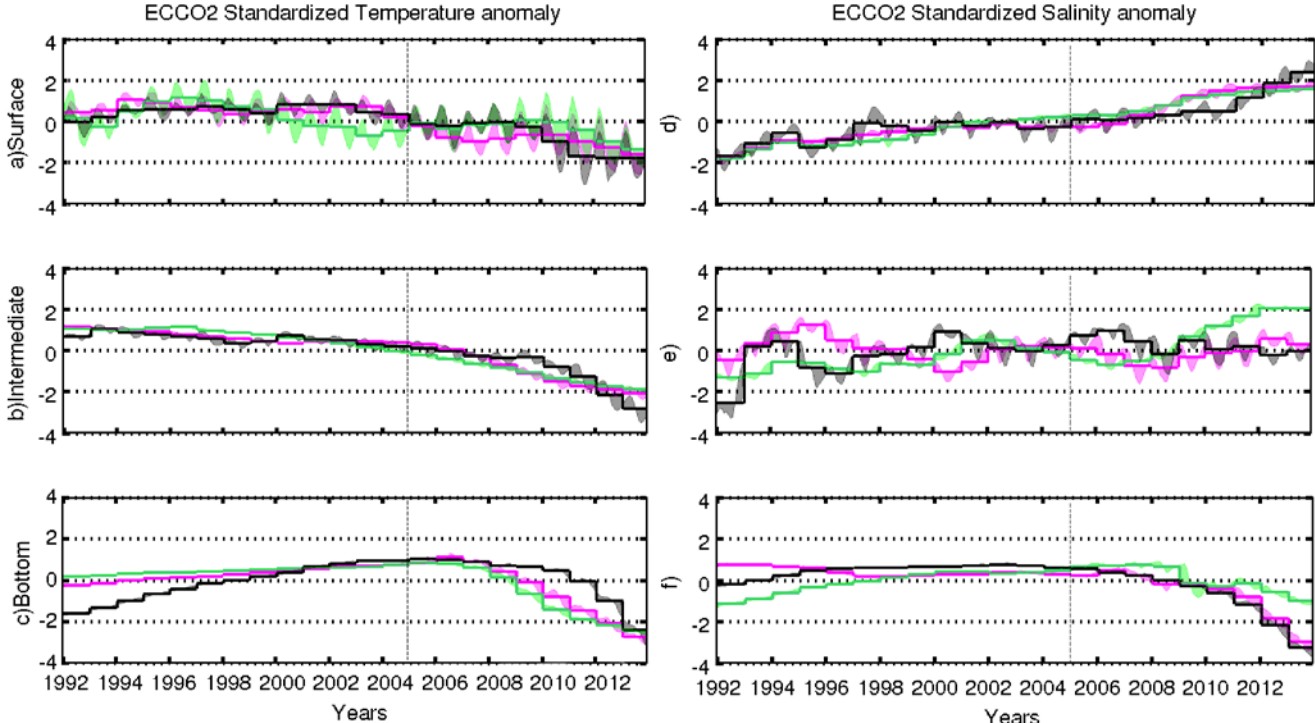

**Figure 8:** Time series of the ECCO2 temperature and salinity normalized anomalies for the Weddell Sea (Green), Indian Ocean (Magenta) and Western Pacific (Gray)sectors. The step plots represent the annual average of the anomalies, while the contours represent the monthly oscillations around the annual average. Dashed vertical line shows the opening of the Weddell Polynya, which stays open until the end of the reanalysis.

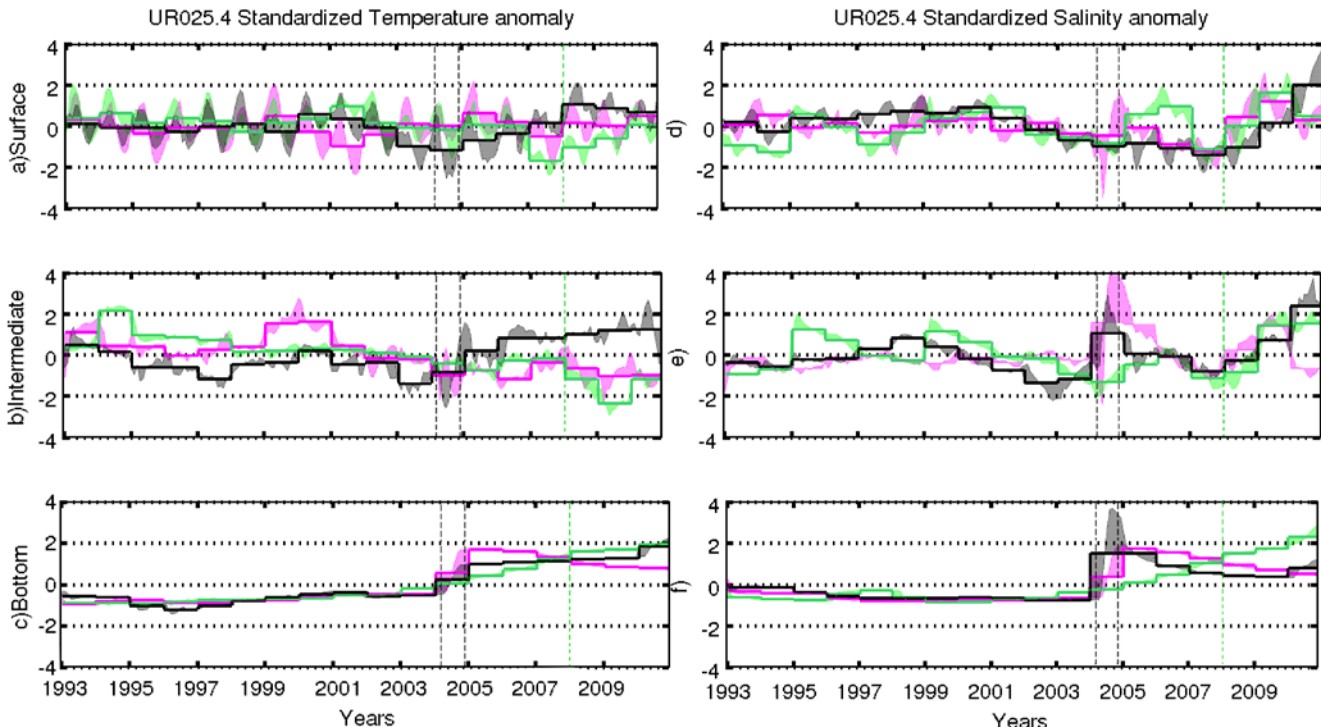

**Figure 9: Same as Figure 8, but for UR025.4. Gray dashed lines show the periods of AABW formation in Indian Ocean and Western Pacific Sectors, while green dashed line show the beginning of AABW formation in Weddell Sea.**

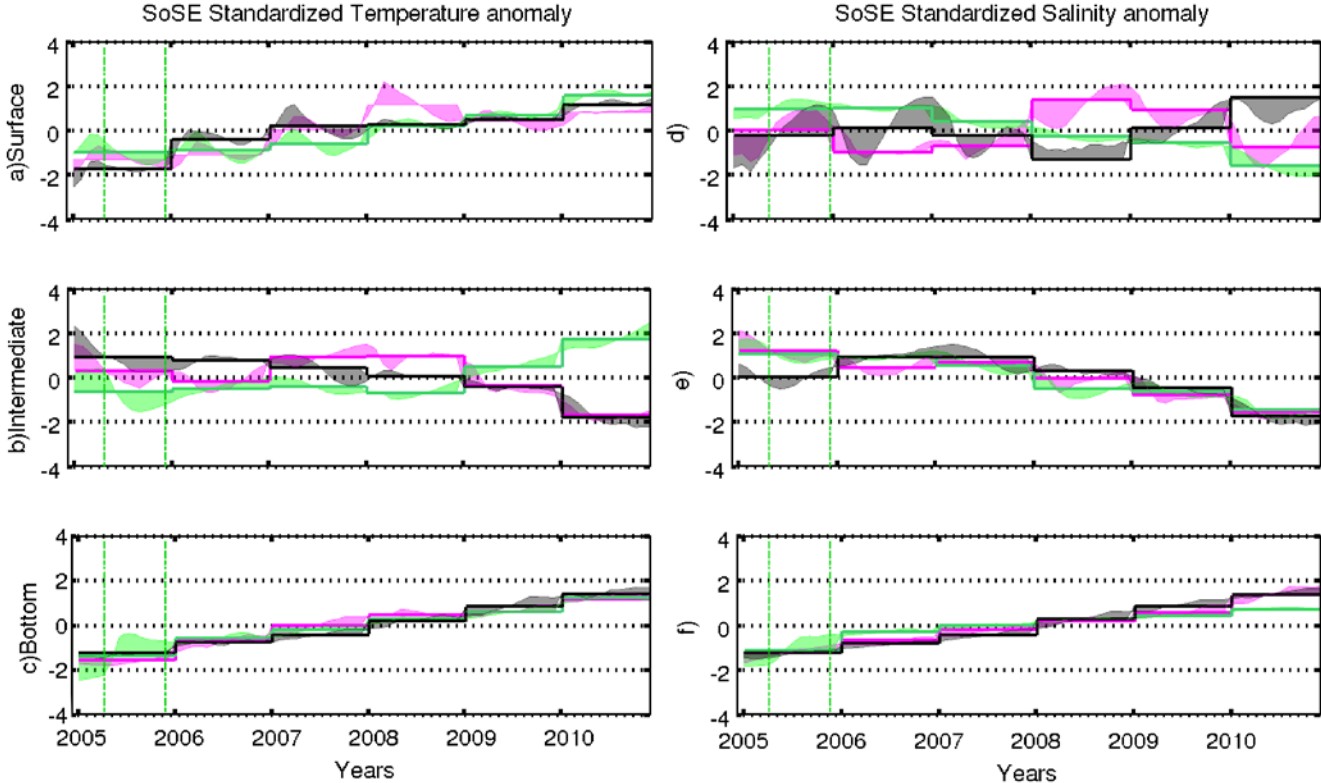

**Figure 10: Same as Figure 8, but for the SoSE reanalysis. Green dashed lines delineate the period in which the polynya stays open in SoSE.**

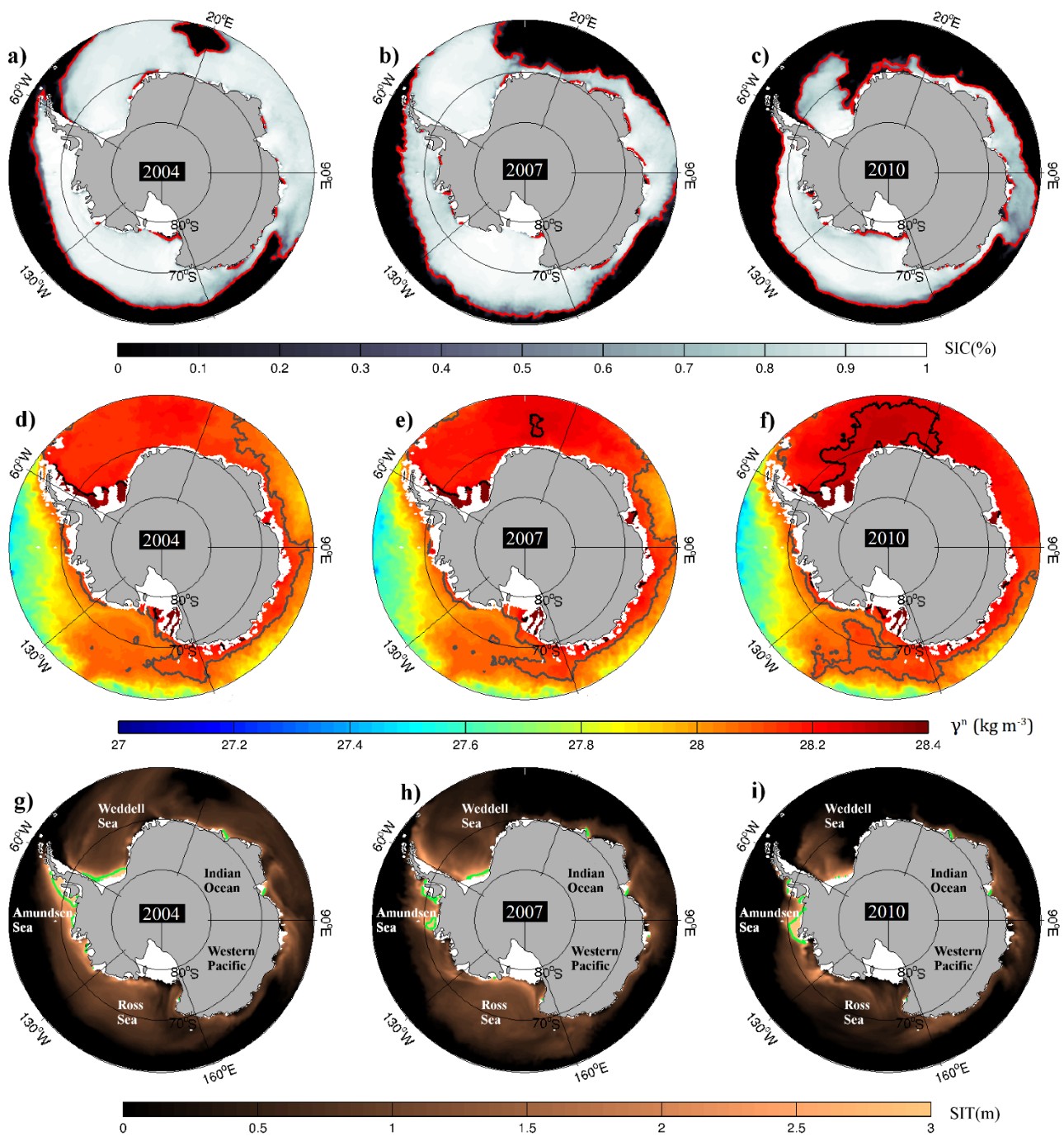

**Figure 11: (a), (b) and (c) are the mean ECCO2 sea ice concentrations in November of 2004, 2007, and 2010, respectively. The red contours delineate the 30% sea ice concentration, which is the border of the polynya. The straight black lines separate each Southern Ocean sector analyzed. (d), (e) and (f) are the mean neutral density filled contours at 700 m for November of 2004, 2007,**

and 2010, respectively. The gray lines delineate the 28.1 kg m$^{-3}$neutral density of WDW and the black lines the 28.27 kg m$^{-3}$ of the WSDW. (g), (h) and (i)are the mean ECCO2 SIT (m)in November of 2004, 2007, and 2010, respectively. The green contours delineate areas with SIT greater than 3.5 m.

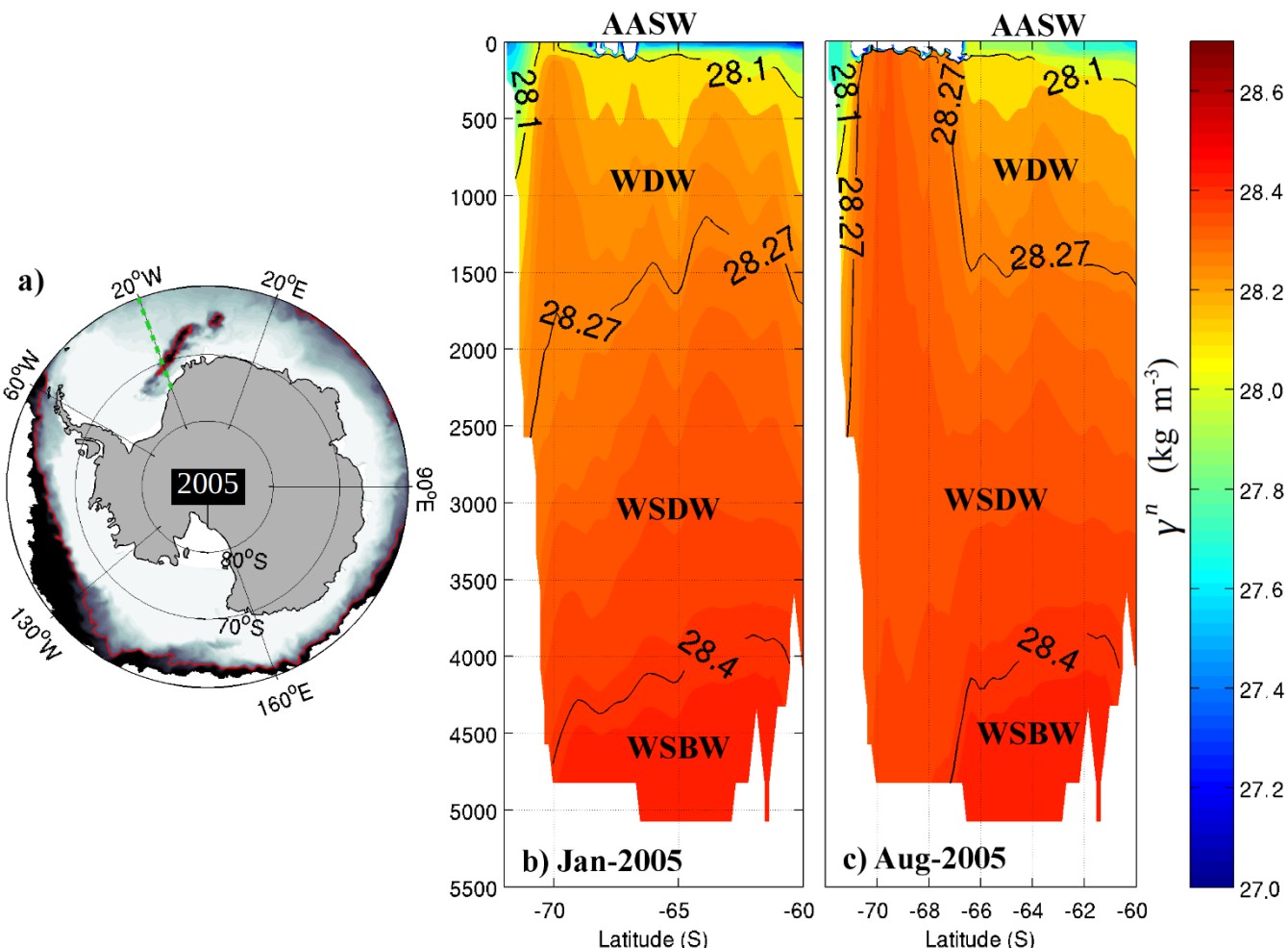

Figure 12:a) A map of the sea ice concentration of SoSE in August 2005 showing the polynya. The transect used is marked by the dashed green line. The black areas are the areas with 0% sea ice concentration. The red line marks the 30% sea ice concentration margin, as the border of the polynya. b) and c) The neutral density contours from a 20° W vertical section in January and August, respectively. The neutral density lines of 28.1 kg m$^{-3}$, 28.27 kg m$^{-3}$ and 28.4 kg m$^{-3}$separate the AASW/WDW,WDW/WSDW and WSDW/WSBW, respectively.

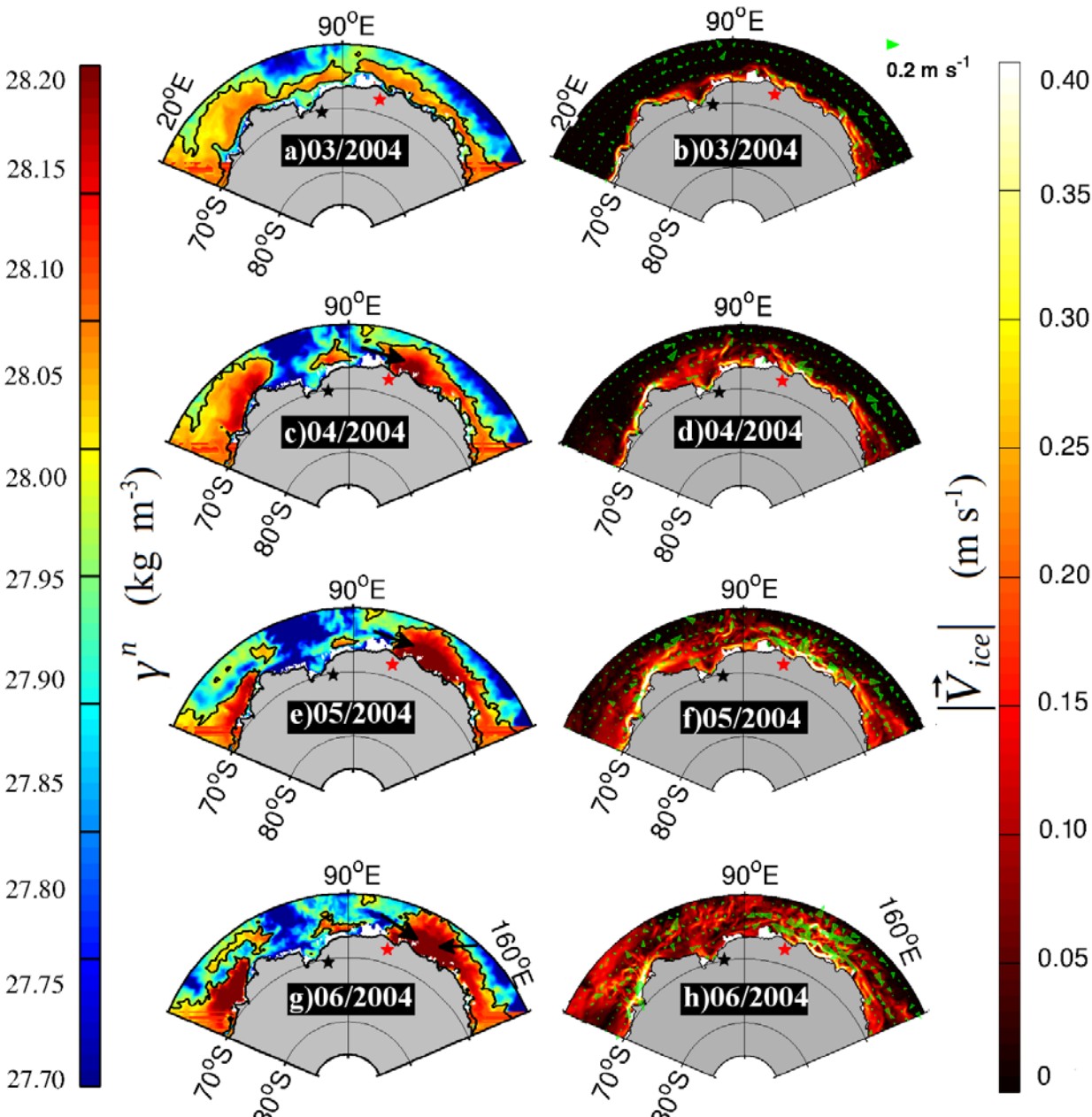

**Figure 13:**UR025.4 neutral densities filled contours at 250 m depth in the center of the UCDW entrainment in the Indian Ocean sector and the Western Pacific sector in March (a), April (c), May (e) and June (f) of 2004.The black lines mark the 28 kg m$^{-3}$ neutral density values that separate UCDW from LCDW. The black arrow represents the direction of the density gradient. Maps (b), (d), (f) and (h) show the sea ice speed module (m/s) for the same months as the neutral density contours. The green arrows show the current speed at 250 m, which is the same depth as the neutral density contours. The black (★)and red stars(★)on all maps represent the locations of Prydz Bay and Vincennes Bay, respectively.

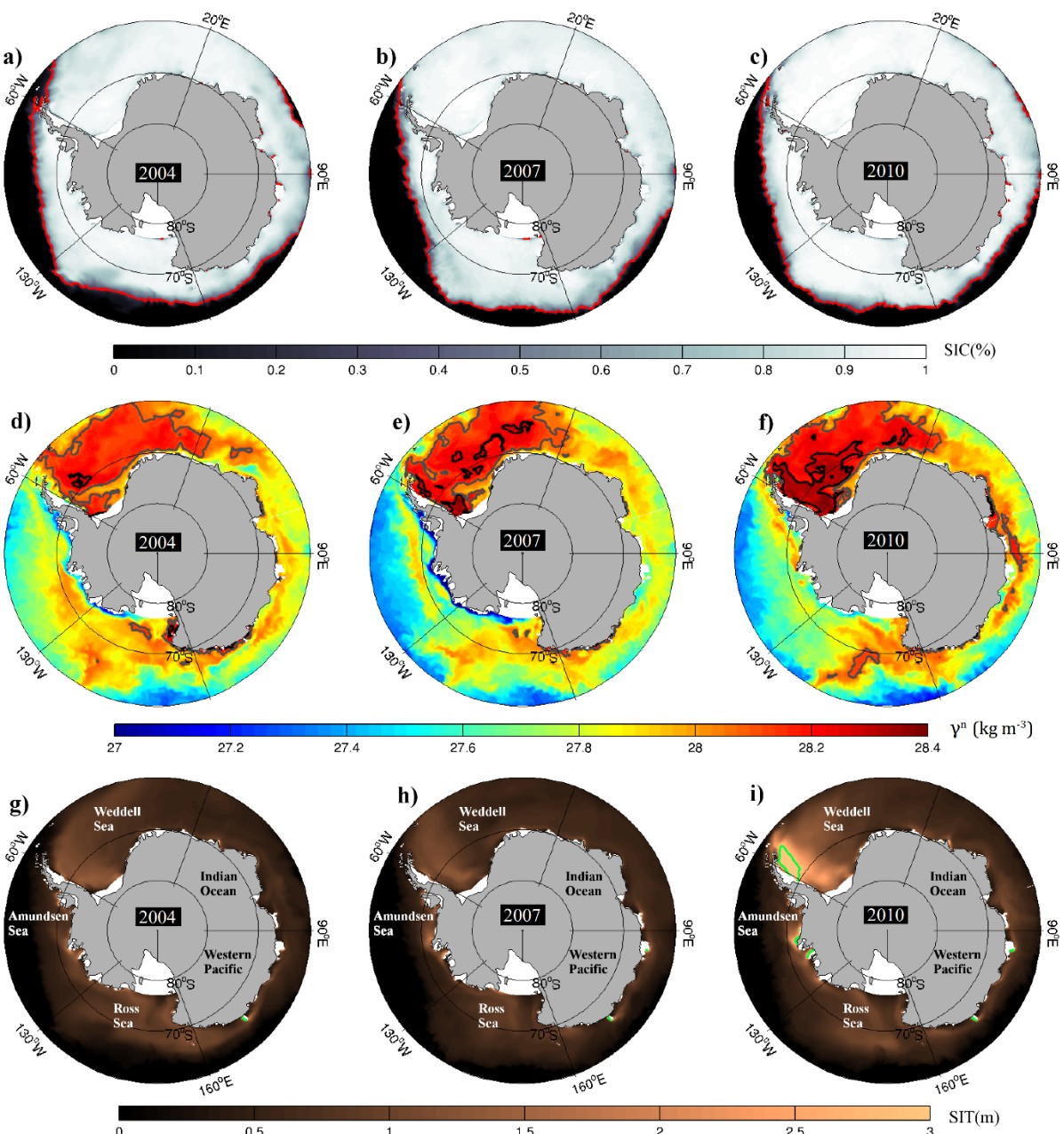

**Figure 14: (a), (b) and (c)The mean UR025.4 sea ice concentration in September of 2004, 2007, 2010, respectively. The red line marks the 30% SIC contour. (d), (e) and (f)The neutral density contours at 700 m depth in September of 2004, 2007, and 2010.The gray lines delineate the 28.1 kg m⁻³neutral density of WDW and the black lines the 28.27 kg m⁻³ of the WSDW. (g), (h) and (i) Monthly sea ice thicknesses of UR025.4 in September of 2004, 2007 and 2010, respectively. The green line marks the 3.5 m sea ice thickness.**