# Peer review of "On deep convection events and Antarctic Bottom Water formation in ocean reanalysis products"

_Ocean Science, 2017_

## Referee Comment (RC1) · Anonymous Referee #1 · 18 Apr 2017

This paper compares water mass formation processes occurring in the Southern Ocean in three different data assimilated ocean model results. This is a very important exercise assessing model performance.

The authors related sea ice to the Weddle Sea Polynya and then the Antarctic Bottom Water. They conducted nice analysis on conversion from a water mass to another due to convection using volume percentage of water masses. They reported that in two of the models (ECCO2 and SoSE) AABW is formed through open ocean convection in Weddell Sea, while in other model (My Ocean University of Reading, UR025.4) through dynamically accurate continental shelf convection and exporting of dense water. I found that these processes are well explained in the text, and relationship between sea ice, open ocean convection and the volumetric percentage is consistent.

[Figure]

The authors argued that the excessive open ocean convection in ECCO2 and SoSE is due to insufficient assimilation of sea ice. I do not agree with the author on this matter. There is no doubt that sea ice is closely related to the open ocean convection, but oceanic processes such as rising of WDW might have initiated open ocean convection and the reduction in sea ice. In addition, what the authors have observed could be initial adjustment. For example, in ECCO2 sudden change occurs in 2004 as explained in the text. How could you show that this is not an adjustment process?

What is causing the differences between the models? I suppose it may be hard to pinpoint the processes causing the differences, but I suggest the authors to compared circulation patterns and vertical profiles of temperature and salinity more carefully. Except for SoSE (Fig. 3), there is no figure showing temperature and salinity. A related question is why UR025.4 performs better. Is this resolve the coastal geometry better? Is it initialized differently?

It must be explained in the references, but I hope there would a description on the assimilated data over the Southern Ocean. Comprehensive explanation on the initialization is necessary. Difference in the initial states might be the cause of the difference in the convection and water mass formation.

In several places, ocean current are introduced while explaining water mass formation. There, however, are no figures and it is not easy to follow the explanation. Please added appropriate figures.

It was concluded that improvements necessary. What kinds of improvement are necessary?

Figures 1, 3 and 4 should be improved. The contours lines except for the black ones in Fig. 4 are hard to see. A scale bar is necessary for SIC.

[Figure]

---

## Referee Comment (RC2) · C. Heuzé (Referee) · 29 Apr 2017

Following the discovery of spurious open ocean deep convection in the Weddell Sea in the ECCO reanalysis two-three years ago, this manuscript investigates whether this behaviour is also found in other reanalysis products, and if so, which mechanism generates it. Reanalyses are often used instead of observations, in data-poor areas such as the Southern Ocean, yet are pretty much models. Such a study is hence vital both for the observational and modelling community.

The manuscript in its present form however does not really answer the questions announced in the abstract. The analysis concentrates on one reanalysis product only, and the explanations lack evidence to back them. The results that are shown are interesting and encouraging, but a substantial amount of rewriting is required. Please

pay attention to the English language as well, and consider asking a native speaker to correct future versions of the manuscript.

Major comment

Section 3 needs to be majorly rewritten so that you properly comment on all three re-analyses and actually demonstrate the mechanisms that you discuss. Try re-organising your section (especially sections 3.2 and 3.3) following this structure:

1. Show a first figure;

2. Comment on it, for all three reanalyses. If they diverge (e.g. SoSe and ECCO are different from UR), start demonstrating the mechanism by showing the next figure; then

2.1 Comment on that next figure for the reanalyses that agree with each other;

2.2 "In contrast, UR..." – Comment on the different reanalysis.

3. Reiterate as many times as necessary until the full process has been demonstrated for all three reanalyses.

Also, make sure that your figures actually show what you are discussing. For example on page 6 from line 28, you use a figure showing year 2004 yet comment on the reanalyses in other years.

Other comments

Throughout the text: Why are some water mass names in italics?

The figures are not consistent. For example, Figs. 1 and 3 feature maps of sea ice concentration, but the third map (Fig. 4) is of sea ice thickness.

Fig. 2 has black lines for "observations", but not Fig. 5 and subsequent figures. Observational water mass distribution/volume should be provided, using the world ocean atlas for example.

P2, line 32: there are more than 15 models in CMIP5 – rephrase as "...found that most

models of the Coupled…" or "…found that all 15 models they studied…"

P3, line 29: this sentence is confusing, what do you mean by "those distinct patterns"? Please rephrase.

P4, lines 17-24: you should summarise the water masses and their densities in a table, that would be clearer.

P5, line 2: are sea ice and ocean currents velocities directly provided, or are they calculated? if so, how?

P6, line 4: where do you show the neutral density layers?

P6, line 20: give the value of the high heat content.

P6, last sentence: that is not true, there is a small region with WSDW in 2004 on Figure 4d.

P7, line 25: "unrealistic percentage" – that joins my previous comment, the reader does not know what would be a realistic value unless you show these in observations too.

Throughout section 3.2: how do you define a significant change? How many percent?

P11, line 26: Thanks for the citation, but that is not really relevant here. Cite rather Kjellsson et al. (2015), doi: 10.1016/j.ocemod.2015.08.003, or Heuzé et al. (2015), doi:10.5194/gmd-8-3119-2015

Figure 8: caption does not say which reanalysis you are showing.

Figs 5,6,7,9,10: present all results for similar water masses with the same vertical range (i.e. same range for all subpanels of surface water, same range for all subpanels with AABW etc)
* * *

---

## Author Comment (AC1) · 8 Jul 2017

Please find enclosed the response to reviewer #1 comments based on the OSD submission doi:10.5194/os-2017-9 and entitled "On the deep convection events and Antarctic Bottom Water formation in ocean reanalysis products" by Aguiar et al. Note that the supplement comprises a complete response letter, and the revised manuscript.

Please also note the supplement to this comment:
https://www.ocean-sci-discuss.net/os-2017-9/os-2017-9-AC1-supplement.zip

---

## Author Response (AR1)

**REPLY LETTER FOR REVIEWER #1**

Anonymous referee

In the following document, the original reviewer comments are in **Bold**. The author's responses are in plain font, and the alterations in the text are in *Italics*.
* * *
**This paper compares water mass formation processes occurring in the Southern Ocean in three different data assimilated ocean model results. This is a very important exercise assessing model performance.**

**The authors related sea ice to the Weddle Sea Polynya and then the Antarctic Bottom Water. They conducted nice analysis on conversion from a water mass to another due to convection using volume percentage of water masses. They reported that in two of the models (ECCO2 and SoSE) AABW is formed through open ocean convection in Weddell Sea, while in other model (My Ocean University of Reading, UR025.4) through dynamically accurate continental shelf convection and exporting of dense water. I found that these processes are well explained in the text, and relationship between sea ice, open ocean convection and the volumetric percentage is consistent.**

We would like to thank the referee for the very encouraging and positive comments on the manuscript! We have thanked both referees in the acknowledgments section:

"*We would finally like to thank C. Heuzé and the anonymous referee for the valuable suggestions that improved the manuscript.*"- Page 19, lines 24-25

Following the suggestions, the manuscript was substantially rewritten to convey the ideas in a more robust storyline. Major text changes were made in section 3, and the results are now described as follows:

Section 3.1 – First, we describe the SIC and SIT alterations for all reanalysis products including their similarities and differences.
Section 3.2- The water mass alterations are discussed by sector and then compared between the reanalysis products. We tried to follow this order of description whenever possible: first ECCO2, then UR025.4 and finally SoSE alterations.
Section 3.3 –An analysis of the temperature and salinity anomalies in the three layers of the Southern Ocean was performed for each model. This analysis provides valuable clues on the mechanisms involved in AABW formation in each model and adds to the discussion of the mechanisms of AABW formation.
Section 3.4 –The results of the previous 3 sections are joined in a detailed unifying explanation, which is explained as many times as needed to convey the main idea to the reader.

Finally, the manuscript had the English carefully revised by the American Journal Experts (AJE), with the following certificate verification key: 2643-6C26-AB4A-DF99-D760

**The authors argued that the excessive open ocean convection in ECCO2 and SoSE is due to insufficient assimilation of sea ice. I do not agree with the author on this matter. There is no doubt that sea ice is closely related to the open ocean convection, but oceanic processes such as rising of WDW might have initiated open ocean convection and the reduction in sea ice.**

We agree with the referee on the fact that both sea ice assimilation and WDW rising played roles in the opening of the polynya. To better convey this idea, some changes were made to the text. In the abstract, the sentence on lines 13 and 14 has been rewritten as follows:

*"We found that two of the products create AABW by open ocean deep convection events in the Weddell Sea that are triggered by the interaction of sea ice with the Warm Deep Water, which shows that the assimilation of sea ice is not enough to avoid the appearance of open ocean polynyas."* - Page 1, lines 13-16

In the discussion of the AABW formation processes in section 3.4, the following has been added:

"*The trigger of the polynya in SoSE is similar to that in ECCO2and was the heat delivery to the surface level by the WDW. The 100-m integrated oceanic heat content calculated is $3.236 \times 10^{22}$ J under the polynya (August 2005), which is higher than the $3.226 \times 10^{22}$ J heat content calculated for August 2008 when there are no ice-free areas. Although the difference is two orders of magnitude lower than the OHC value, the difference results in a one degree warmer surface temperature in August 2005 than in August 2008 and crosses the freezing point of seawater."* – Page 14, Lines 11-15

In the summary and conclusions section, the following sentence has been added:

*"Furthermore, weak stratification that enhanced WDW heat release to the surface seemed to be one of the main triggers of the Weddell Sea Polynya opening in the ECCO2 and SOSE reanalysis products.*" - Page 18, Lines 19-20

**In addition, what the authors have observed could be initial adjustment. For example, in ECCO2 sudden change occurs in 2004 as explained in the text. How could you show that this is not an adjustment process?**

We agree that the polynya opening in the Southern Ocean simulations can generally be due to the adjustments to the initial settings of the model. However, in ECCO2, the polynya opens after twelve years of simulations, and it only increases after opening, so we do not believe the establishment of the polynya in ECCO2 is an expression of an initial adjustment. That is also in agreement with the findings of Azaneu et al. (2014). In the case of SoSE, a one-year spin-up time is applied in 2004 to adjust the reanalysis to its initial settings. The SoSE output is then released only for the period after the spin-up procedure, and hence it is considered that the solution is already adjusted to its initial settings. Therefore, we do not believe the polynya opening in this reanalysis is an adjustment process. Some sentences were added to the text to describe this point of discussion:

"*Finally, although the polynya in SoSE occurs at the beginning of the reanalysis output, we do not believe its opening is a result of an initial adjustment process, since a one year spin-up procedure is conducted in the prior year (2004) to bring the SoSE to its equilibrium conditions…*" – Page 15, Lines 8-10

**What is causing the differences between the models? I suppose it may be hard to pinpoint the processes causing the differences, but I suggest the authors to compared circulation patterns and vertical profiles of temperature and salinity more carefully. Except for SoSE (Fig. 3), there is no figure showing temperature and salinity. A related question is why UR025.4 performs better. Is this resolve the coastal geometry better? Is it initialized differently?**

We agree with the referee that a better description of the causes of the different hydrographic and ocean dynamic patterns in ECCO2, SoSE and UR025.4 needed to be further explored. A whole section (Section 3.3) discussing the temperature and salinity changes in the surface, intermediate and bottom layers was added to better explain

the differences that have led to the AABW formation in each model. The section regarding temperature and salinity values can be found in the revised manuscript.

By analyzing the temperature and salinity patterns, we found that water column stratification might be one of the main reasons why ECCO2 experiences an open ocean polynya and UR025.4 does not. The ECCO2 bottom layer experiences warming prior to the polynya opening, while the intermediate layer seems to cool down. Those changes might have led to a less stratified water column, allowing vertical heat transfer to melt sea ice and open the ECCO2 polynya in the Weddell Sea.

To expand upon those ideas, some other sentences were inserted in the main text. In the new section 3.3, we added the following:

*"Cooling and salinity increase in both the surface and intermediate layers of the Weddell Sea sector before 2006 (Figure 8b,e) when considered together with the continuous warming in the bottom layer (Figure 8c), reveal an important feature since it allows for vertical stratification to weaken, thus favoring deep convection."* – Page 11, Lines 31-32

Additionally, in section 3.4 we discussed the possible reasons why UR025.4 performs better, while ECCO2 and SoSE create spurious open ocean polynyas:

*"Finally, in all three reanalysis products investigated in this study, the AABW formation occurred due to a higher content of warm CDW-derived waters and interaction with sea ice. Why then the mechanism of AABW formation in UR025.4 is different from the other two reanalysis (ECCO2 and SoSE)? One of the possible explanations might be that the advection of CDW-derived waters in UR025.4 originates from the east in the Weddell Sea. There is a region with consistently low sea ice concentrations and thicknesses near the center of the Weddell Sea, which is due to the natural isopycnal uplift inside the Weddell Gyre (de Steur et al., 2007). Hence, the warm CDW waters that flow west along the isopycnals tend to rise when they reach the central Weddell Gyre, while they stay roughly at the same depths when they flow east towards the Indian Ocean sector and only upwell along the coast due to coastal divergence. Thus, the warm water in the deep Weddell Sea layers is expected to exchange heat with the sea ice in the central Weddell Gyre, which can likely lead to the establishment of a polynya. In fact, Timmermann and Beckmann (2004) attempted to accurately reproduce that so-called warm water halo and found an enhanced vertical heat exchange with sea ice, which resulted in the opening of an oceanic polynya in the Weddell Sea. In addition, long-term cooling of the intermediate layers and the warming in the bottom layers of ECCO2 might have played a role in polynya establishment. Those trends decrease Southern Ocean vertical stratification and allow heat to be transported upwards and deep convection to happen. Azaneu et al. (2014) discussed the possible triggers of the ECCO2 polynya and suggested that the long-term warming of the bottom waters was one of the main factors that contributed to the polynya establishment and subsequent expansion. In addition, both ECCO2 and SoSE use the same ocean model and similar modeling frameworks, so we cannot rule out the appearance of the polynya in both models as an expression of similar model features of the reanalysis products."* – Page 17, Lines 17-34

**It must be explained in the references, but I hope there would a description on the assimilated data over the Southern Ocean. Comprehensive explanation on the initialization is necessary. Difference in the initial states might be the cause of the difference in the convection and water mass formation.**

A more detailed description of the assimilated data and the initial conditions in each reanalysis product was added to the manuscript. In the manuscript, while describing the ECCO2 framework, the following paragraph has been added:

"*The data assimilated by ECCO2 includes temperature and salinity profiles from the World Ocean Circulation*

*Experiment database, Argo floats, and XBT measurements; sea surface temperature measurements from the Group of High Resolution Sea Surface Temperature (GHRSST); sea level anomaly data from altimetry; temporal mean sea levels from Maximenko and Niiler (2005); sea ice concentrations from passive microwave data; sea ice thickness from Upward Looking Sonar; and finally sea ice motion from the QuikSCAT and RADARSAT-GPS*
5   *radiometers. A Green's function method is used to calibrate the control variables (Menemenlis et al., 2005) and the initial parameters, which include initial temperature and salinity conditions; background vertical diffusivity; atmospheric surface boundary conditions; critical Richardson numbers; air-ocean, ice-ocean and air-ice drag coefficients; albedo coefficients of ice, ocean and snow; bottom drag and vertical viscosity. ECCO2 is run directly from its initial conditions, without the use of a spin-up period to bring the model to equilibrium (Aksenov et al.,*
10   *2016).*" – Page 3, Lines 16-25.

The data constraints of SoSE were added as follows:

"*The data constraints of SoSE include temperature and salinity fields from Argo floats and instrument-mounted*
15   *elephant seal profiles; CTD and XBT profiles from the Scripps Institution of Oceanography High Resolution CTD/XBT network and the CliVar and Carbon Hydrographic Data Office; sea surface height from the Radar Altimetry Database System; sea surface temperature from microwave radiometers; sea ice concentrations from the National Snow and Ice Data Center; mean dynamical topography from the Technical University of Denmark; and bottom pressure estimates from the ECCO project. The other measurements used in the assimilations were*
20   *taken from the Antarctic Marine Living Resources Program, the Long-Term Ecological Research Network and the Diapycnal and Isopycnal Mixing Experiment in the Southern Ocean. The SoSE initialization includes a one-year spin up period using the dataset from the 2004Ocean Comprehensible Atlas (OCCA - Forget, 2010) with adjusted kinetic energy. The optimization method applied in the SoSE changes the initial temperatures and salinities, and a one-week adjustment shock occurs when the model begins to run. Furthermore, neither the OCCA nor the*
25   *SoSE were optimized to eliminate spurious drifts (M. Mazloff, personal communication).*" – Page 4, Lines 12-23.

The data assimilated by the UR025.4 was also added:

"*UR025.4 data includes temperature and salinity profiles from the EN3 dataset, including Argo floats, XBT, CTD,*
30   *TAO and PIRATA measurements; sea surface temperature and altimetry data from the International Comprehensive Ocean-Atmosphere Data Set; and sea ice concentration from the Ocean and Sea Ice Satellite Application Facility.UR025.4 uses initial conditions of EN3 climatology to start the simulation. Authors considered that the 3d-Assimilation scheme allowed fast adjustment of surface and subsurface properties, and hence no spin up period is used in this reanalysis (Valdivieso et al., 2014)*" Page 4, Line 3-8.
35

**In several places, ocean current are introduced while explaining water mass formation. There, however, are no figures and it is not easy to follow the explanation. Please added appropriate figures.**

We agree that a figure showing the acceleration of zonal velocities around Prydz Bay is necessary since our
40   main conclusions of UR025.4 are based on the enhanced sea ice transport due to the enhanced current velocities in that area. To show that zonal velocity increases in Prydz Bay, we calculated the mean zonal speed profile along 70.125°W between the coast and 64°S. This figure can be found in the supplementary material (Figure S1). It is possible to see in Figure S1 that after 2004, the negative zonal velocities increase especially at the surface. Those strong negative current velocities show the intensification of the westward flow of the Antarctic
45   Coastal Current (ACoC) around Prydz Bay, which is the main velocity alteration that occurs during AABW formation.

[Figure]

**Figure S1. The temporal evolution of a mean current speed profile in UR025.4. The red line at the top shows the transect used to calculate the mean profile (70.125°W)**

In addition, the green arrows in Figure 13 show the intensity and direction of the current at 250 m depth, which indicates the intensification of the ACoC and the offshore-directed buoyancy current.

**It was concluded that improvements necessary. What kinds of improvement are necessary?**

Based on the dynamics of ECCO2 and SoSE, we believe that improvements in the parameters controlling heat exchange between the sea ice and surface water are required. In parallel, understanding the causes of bottom layer warming and intermediate layer cooling in ECCO2 is necessary for future studies to better delineate the mechanisms generating stratification. The following text has been added to the section 4 to convey those ideas:

*"Furthermore, weak stratification that enhanced WDW heat release to the surface seemed to be one of the main triggers of the Weddell Sea Polynya opening in the ECCO2 and SOSE reanalysis products. The WDW increase reported here is consistent with the observed results reported by Kerr et al. (2017 - under review), who found a consistent increase of the WDW contribution to the total mixture of deep and bottom waters in the Weddell Sea from 1984 to 2014, despite the high degree of interannual variability. However, since no real open ocean polynya has been reported since the 1970s (Gordon 1978), a critical analysis of the model mechanisms of heat exchange between the surface waters and sea ice is required in the future to efficiently understand the role of WDW in open ocean polynya establishment. In addition, since bottom layer warming and intermediate layer cooling are the possible mechanisms that diminished stratification in ECCO2, further evaluation of the causes of those trends is needed to understand the primary factors leading to the weak ocean surface stratification."* – Page 18, Lines 19-27.

**Figures 1, 3 and 4 should be improved. The contours lines except for the black ones in Fig. 4 are hard to see. A scale bar is necessary for SIC**

As suggested, the quality of the figures was enhanced. The white contour lines were changed to gray lines, and a scale bar was added for the SIC values. Please also notice that the ECCO2 and UR025.4 figures now display both SIC and SIT. Additionally, the following changes were made to the figures:
- Previous Figure 1 is now Figure 11

- Previous Figure 3 is now Figure 12
- Previous Figure 4 is now Figure 14

Those figures are shown below:

[Figure]

*Figure 11: (a), (b) and (c) are the mean ECCO2 sea ice concentrations in November of 2004, 2007, and 2010 respectively. The red contours delineate the 30% sea ice concentration, which is the border of the polynya. The straight black lines separate each Southern Ocean sectors analyzed. (d), (e) and (f) are the mean neutral density filled contours at 700 m for November of 2004, 2007, and 2010, respectively. The gray lines delineate the 28.1 kg m-3neutral density of WDW and the black lines the 28.27 kg m-3 of the WSDW. (g),*
10 *(h) and (i) are the mean ECCO2 SIT (m) in November 2004, 2007, and 2010, respectively. The green contours delineate areas with SIT greater than 3.5 m.*

[Figure]

*Figure 12: a)A map of the sea ice concentration of SoSE in August 2005 showing the polynya. The transect used is marked by the dashed green line. The black areas are the areas with 0% sea ice concentration. The red line marks the 30% sea ice concentration margin, as the border of the polynya. b) and c) The neutral density contours from a 20 W vertical section in January and August, respectively. The neutral density lines of 28.1 kg m-3, 28.27 kg m-3 and 28.4 kg m$^{-3}$separate the AASW/WDW, WDW/WSDW and WSDW/WSBW, respectively.*

[Figure]

*Figure 14: (a), (b) and (c)The mean UR025.4 sea ice concentration in September of 2004, 2007, 2010, respectively. The red line marks the 30% SIC contour. (d), (e) and (f)The neutral density contours at 700 m depth in September of 2004, 2007, and 2010. The gray lines delineate the 28.1 kg m⁻³neutral density of WDW and the black lines the 28.27 kg m⁻³ of the WSDW. (g), (h) and (i)Monthly sea ice thicknesses of UR025.4 in September of 2004, 2007 and 2010, respectively. The green line marks the 3.5 m sea ice thickness.*

**REPLY LETTER FOR REFEREE #2**

Reviewer: Celine Heuzé (celine.heuze@gu.se)

In the following document, the original reviewer comments are in **Bold**. The author's responses are in plain font,

5 and the alterations in the text are in *Italics*.
* * *
**Following the discovery of spurious open ocean deep convection in the Weddell Sea in the ECCO reanalysis two-three years ago, this manuscript investigates whether this behavior is also found in other**
10 **reanalysis products, and if so, which mechanism generates it. Reanalysis are often used instead of observations, in data-poor areas such as the Southern Ocean, yet are pretty much models. Such a study is hence vital both for the observational and modelling community.**
**The manuscript in its present form however does not really answer the questions announced in the abstract. The analysis concentrates on one reanalysis product only, and the explanations lack evidence**
15 **to back them. The results that are shown are interesting and encouraging, but a substantial amount of rewriting is required. Please pay attention to the English language as well, and consider asking a native speaker to correct future versions of the manuscript.**

We would like to thank the reviewer for the detailed and valuable comments on the manuscript. We have thanked
20 both referees in the acknowledgments section.

"*We would finally like to thank C. Heuzé and the anonymous referee for the valuable suggestions that improved the manuscript.*"- Page 19, lines 24-25

25 Following the suggestion, the manuscript was reorganized, and several sections were rewritten to clarify the mechanisms involved in AABW formation. We agree that the analysis mainly focused on the UR025.4 reanalysis product. Therefore, we extended the explanations and discussions of the two other products (SoSE and ECCO2). New figures were added, and some of the old figures were edited to better support the explanations. A new results section was added to discuss the temperature and salinity changes and their links to brine release and
30 surface cooling during AABW formation in the three models. We believe this section was a piece that was missing from the original manuscript and was necessary to back up the explanations of the AABW formation mechanisms. Finally, the authors also agree that the question proposed in the abstract was somewhat confusing, so this question was rewritten as follows:

35 *"Despite those events are well described in non-assimilatory ocean simulations, the recent appearance of a massive open-ocean polynyas in the Estimating the Circulation and Climate of the Ocean Phase II reanalysis product (ECCO2) raises questions on which mechanisms are responsible for those spurious events and if they are also present in other state-of-the-art reanalysis products."*- Page 1, lines 9-12.

40 Finally, the manuscript had the English carefully revised by the American Journal Experts (AJE), with the following certificate verification key: 2643-6C26-AB4A-DF99-D760

45
**Major comment**

**Section 3 needs to be majorly rewritten so that you properly comment on all three reanalyses and actually demonstrate the mechanisms that you discuss. Try re-organizing your section (especially sections 3.2 and 3.3) following this structure:**
**1. Show a first figure;**
**2. Comment on it, for all three reanalyses. If they diverge (e.g. , SoSe and ECCO are different from UR), start demonstrating the mechanism by showing the next figure; then**
**2.1 Comment on that next figure for the reanalyses that agree with each other;**
**2.2 "In contrast, UR**
**...**
**" – Comment on the different reanalysis.**
**3. Reiterate as many times as necessary until the full process has been demonstrated for all three reanalyses.**

We would like to thank the referee for the recommendation. Following the suggestion, the results section was rewritten to convey the ideas in a clearer manner. Specifically, section 3.2 was substantially reorganized to strictly discuss the water mass alterations. In section 3.4 (previously section 3.3), the previous results were combined to explain how AABW formed in the models. Now, the results section proceeds as follows:

Section 3.1 – First, we describe the SIC and SIT alterations for all reanalysis products including their similarities and differences.
Section 3.2- The water mass alterations are discussed by sector and then compared between the reanalysis products. We tried to follow this order of description whenever possible: first ECCO2, then UR025.4 and finally SoSE alterations.
Section 3.3- An analysis of the temperature and salinity anomalies in the three layers of the Southern Ocean was performed for each model. This analysis provides valuable clues on the mechanisms involved in AABW formation in each model and adds up to the discussion of the mechanisms of AABW formation. The analysis is performed for each model separately to avoid confusion.
Section 3.4 –The results of the previous 3 sections are joined in a detailed unifying explanation, which is explained as many times as needed to convey the main idea to the reader.

**Also, make sure that your figures actually show what you are discussing. For example on page 6 from line 28, y, or use a figure showing year 2004 yet comment on the reanalyses in other years.**

The appropriate figure addressing in this discussion was included. The previous sentence was "In UR025.4 a different process occurs. After 2005, SIT in eastern Antarctic Peninsula rise, reaching values higher than three meters in 2009, and only then starting to decrease (Figure 4a-c)" and referenced the maps of sea ice thickness. Now, the sentence is rewritten in a paragraph that references the annual mean sea ice thickness values as follows:

"*Conversely, UR025.4 exhibits annual SIT increases in the Weddell Sea, almost doubling that signaling 2009 (Figure 2b)." – Page 7, Lines 11-12*.

Additionally, we have checked all cited Figures regarding the companion sentences to ensure a clear understanding and fluency throughout the text.

**Other comments**
**Throughout the text: Why are some water mass names in italics?**

The water mass names were in italics in the first submitted version to highlight the first time that the water mass name appeared in the text. Since this generated confusion, the italics were removed.

**The figures are not consistent. For example, Figs. 1 and 3 feature maps of sea ice concentration, but the third map (Fig. 4) is of sea ice thickness.**

We agree that the use of different sea ice variables when analyzing polynya establishment is not adequate. According to the reviewer suggestions, the following changes were made to the figures:

- Sea ice thickness maps were added to the ECCO2 snapshots (previous Figure 1, current Figure 11)
- Sea ice concentration maps were added to the UR025.4 snapshots (previous Figure 4, current Figure 14).

Since we do not infer any additional information from the SIT maps of SoSE, the figure showing SoSE polynya was not changed.

**Fig. 2 has black lines for "observations", but not Fig.5 and subsequent figures. Observational water mass distribution/volume should be provided, using the world ocean atlas for example.**

We agree with the reviewer on the fact that accessing real water mass volumes is important to determine whether or not the modeled water mass volumes represented in the reanalysis are realistic. However, the current database of hydrographic variables (temperature and salinity) in the Southern Ocean is not detailed enough to accurately calculate the monthly variation of water mass volumes in each sector, as was done for the reanalysis products. A sentence explaining this issue was added to the text, as will be shown. Some studies, however, pinpoint mean percentages of water masses in parts of the Southern Ocean, and they were added to the discussion of the water mass percentages. In Section 3.2 was added:

*"Tomczak and Liefrink (2005) analyzed the mean AABW contribution in the Western Pacific sector using ocean observations from the SR03 World Ocean Circulation Experiment transect (between 130°E and 150°E, and from 44°S to66°S). The study found that AABW fills approximately 30% of the sector, a percentage lower than the 43% found in ECCO2 in 2012."* – Page 9, Lines 25-28.

Also in Section 3.4 was added:

"*Due to limited data sampling, real ocean monthly estimates of WSBW variability are not currently possible. However, some efforts have been made by previous studies to account for the average contribution of WSBW to the Weddell Sea sector. Pardo et al., (2012) used extended Optimum Multiparameter Analysis (eOMP) to quantify the volumes of the Southern Ocean water masses and found that the longitudinal limits of our Weddell Sea sector was filled with approximately 26±0.2% of WSBW, a percentage substantially lower than the 70% of WSBW estimated by ECCO2 in 2013. This previous article uses 45°S as the northern limit for the volume calculations, while our calculation uses 60°S, which accounts for some of the difference in the volume values.*" – Page 13, Lines 11-17.

**P2, line 32: there are more than 15 models in CMIP5 – rephrase as "...found that most models of the Coupled..." or "...found that all 15 models they studied..."**

This part was rewritten as follows:

"*Additionally, Heuzé et al. (2013) found that most models of the Coupled Model Intercomparison Project (CMIP) Phase 5 failed to represent the formations of dense waters accurately and instead created AABW by open ocean deep convection.*". – Page 1,Lines 33-34 and Page 2, Line 1.

**P3, line 29: this sentence is confusing, what do you mean by "those distinct patterns"? Please rephrase.**

The sentence was rephrased as *"The distinct simulation characteristics of the reanalysis products, such as the initialization methods and the assimilated variables, help track how the different features in the simulation frameworks affect AABW production.*" – Page 4, Lines 25-26.

**P4, lines 17-24: you should summarize the water masses and their densities in a table, that would be clearer.**

We agree with the referee in that matter, and hence an additional table (Table 1) was added to summarize the water mass densities for the Southern Ocean.

**P5, line 2: are sea ice and ocean currents velocities directly provided, or are they calculated? if so, how?**

The zonal and meridional components of the sea ice velocity are directly provided by the reanalysis, as well as the current velocities. This information was added to the main text:

*"Finally, for a better description of the AABW formation process in UR025.4, we included analyses of the sea ice and ocean currents, all of which were provided by the reanalysis product."* – Page 6, Lines 10-11.

**P6, line 4: where do you show the neutral density layers?**

That sentence was miswritten in the manuscript since we do not show neutral density layers, but neutral density contour maps. With the rewriting, that part was relocated to section 3.4 and was rewritten as follows:
"*The anomalous signals identified by the average SIC and SIT distribution in ECCO2 are mainly connected to the appearance of a large-scale sensible heat polynya the Weddell Sea sector (Figure 11a-c) and the neutral density alterations (Figure 11d-f), as previously pointed out by Azaneu et al. (2014).*" – Page12, Lines 24-26.

**P6, line 20: give the value of the high heat content.**

We agree that the value of the heat content is important for the discussion. This part of the manuscript now reads as follows:

"*The 100-m integrated oceanic heat content calculated is $3.236 \times 10^{22}$ J under the polynya (August 2005), which is higher than the $3.226 \times 10^{22}$ J heat content calculated for August 2008 when there are no ice-free areas. Although the difference is two orders of magnitude lower than the OHC value, the difference results in a one degree warmer surface temperature in August 2005 than in August 2008 and crosses the freezing point of seawater. Different from ECCO2, WDW in SoSE is present at the surface before the winter (Figure 10b). With the advancement of the sea ice in the winter of 2005, the WDW enduring high heat content at the surface delays sea ice formation until December, and as a result, an elongated polynya occurs in the Weddell Sea.*" – Page 14, Lines 12-18

**P6, last sentence: that is not true, there is a small region with WSDW in 2004 on Figure 4d.**

Thank you. The sentence was rewritten as follows: "*Before the thickening event, WSDW is present at approximately 700 m only in a small region east of the Antarctic Peninsula, while WDW takes up the majority of the Weddell Sea (Figure 14d).*" – Page 16, Lines 12-14.

**P7, line 25: "unrealistic percentage" – that joins my previous comment, the reader does not know what would be a realistic value unless you show these in observations too. Throughout section 3.2: how do you define a significant change? How many percent?**

Similar to the answer for the comment on Figure 2, monthly water mass volume variability estimates in the Southern Ocean are not possible today due to low data cover. Therefore, to understand whether or not a water mass volume is high, we compared the modeled water volumes with the mean volumes estimated by previous studies. We consider a "significant change" to be any percentage that is higher than the mean + one standard deviation of the real ocean water mass percentages. Some of the real ocean water mass estimates do not offer standard deviations, and hence we considered a change significant only based on the visual analysis of the oscillations in the water mass time series. Given those rules, the following text was added to clarify when the percentages are higher than expected:

 "*Due to limited data sampling, real ocean monthly estimates of WSBW variability are not currently possible. However, some efforts have been made by previous studies to account for the average contribution of WSBW to the Weddell Sea sector. Pardo et al., (2012) used extended Optimum Multiparameter Analysis (eOMP) to quantify the volumes of the Southern Ocean water masses and found that the longitudinal limits of our Weddell Sea sector was filled with approximately 26±0.2% of WSBW, a percentage substantially lower than the 70% of WSBW estimated by ECCO2 in 2013.*" Page 13, Lines 11-17

As described in the comment of Figure 2, in section 3.2, the following comparisons with real ocean data were added to the discussion of Weddell Sea sector water mass volumes:

"*Pardo et al. (2012) evaluated the mean volume of deep and bottom waters below 45°S and found that the Weddell Sea water column was comprised of approximately 25±8% of NADW. Within the Weddell Sea, NADW is transformed, and part of it becomes WDW after entering the Weddell Gyre (Carmack, 1974); hence, the 36% value of WDW in ECCO2 is an overestimation, because it surpasses the total percentage of its more widely distributed source water (NADW).*" – Page 7, Lines 31-32 and Page 8, Lines 1-3.

While discussing the Western Pacific water masses in the same section, the following was added:

"*Tomczak and Liefrink (2005) analyzed the mean AABW contribution in the Western Pacific sector using ocean observations from the SR03 World Ocean Circulation Experiment transect (between 130°E and 150°E, and from 44°S to66°S). The study found that AABW fills approximately 30% of the sector, a percentage lower than the 43% found in ECCO2 in 2012.*" – Page 9, Lines 25-28

**P11, line 26: Thanks for the citation, but that is not really relevant here. Cite rather Kjellsson et al. (2015), doi: 10.1016/j.ocemod.2015.08.003, or Heuzé et al. (2015), doi:10.5194/gmd-8-3119-2015**

Thank you. The citation was corrected to refer to Heuzé et al. (2015).

**Figure 8: caption does not say which reanalysis you are showing.**

The proper model identification was added to the caption.

**Figs 5,6,7,9,10: present all results for similar water masses with the same vertical range(i.e. , same range for all subpanels of surface water, same range for all subpanels with AABW**

All graphs of water mass volume percentages were edited to present the same vertical axis length as suggested.

---

## Editor Decision (ED1)

Review of Aguiar et al., first revision

The manuscript has gained a lot in clarity, but part of the narrative is still confusing and does not seem to hold. In particular, from section 3.1 to 3.2, then 3.2 to 3.3, the years where the described events start seem to change. I suggest changes in this document in order to clarify the chronology of the events and verify that the proposed mechanisms are valid.

**Major changes**

For each reanalysis, please indicate

- In the text at the beginning of section 3.2, the dates of the polynyas given in section 3.1;
- On figures 3 to 7, these same dates should be represented as vertical bars with different colours and/or line styles, for example.
- In the text at the beginning of section 3.3, the dates at which AABW formation occurs in each sea inferred from figures 3 to 7;
- On figures 8 to 10, again potentially as vertical bars or shaded rectangles of different colours for the different regions.

Then throughout the text, check that you do describe what happens at those dates that are specific to each region and each reanalysis. For example page 11 line 18, you explain that the Weddell Sea sector does not show the same changes as the other two regions in 2004 in UR025.4. That is not surprising, since as you write page 8 lines 22-23, there WSBW is formed only in 2008 and 2009! Likewise, page 12 you write that WSBW formation occurs "in the first few months of the time series" for SoSE, yet your description page 8 mentions WBSW pulses later, in 2006 and 2007.

**Minor changes**

Throughout the text: despite the language check, there are a lot of typos, blank spaces or letters missing.

P1, l9: Change "despite" into "although".

P1, l23: "long_er_ and _more_ elegant" than what?

P3: Potentially split section 2 into two subsections, one about "Reanalysis and observational products" starting line 10, and one about "Methods" starting P5 l7.

P4, l25-26: This sentence has no clear link with the one after. Move it somewhere else, or add some transition.

P5, l32: "their links with the processes being evaluated" – reference needed.

P7, l14: mostly in the Weddell Sea.

P8, l12 and l20: this comment is valid throughout the manuscript and **was already made during the previous round of reviews**: give the actual values! Don't just write that it increases

P8, l22: for clarity, write ECCO2 instead of "the previous reanalysis product".

P8, l25: the behaviour of SoSE is similar to that of ECCO2 only until 2008. Please discuss what happens from 2008 onwards.

P10, l13: see major comment, give clearly "the periods and locations".

P10, l27-28: no, it is not apparent that cooling was the main mechanism. You do not show it, and you even soften your argument several lines later by writing that cooling could be "favouring deep convection". I'd remove the sentence l27-28 and keep the ones after.

P11, l12: what do you mean by "consistent with the changes in temperature"? That it happens simultaneously? That the increase in salinity and temperature makes sense you think (if so, explain why)?

P11, l24 onwards: not proven. You need to talk more about the timing of the events, how the salinity anomaly would propagate from the surface down.

P13, l4-6: where do you show the properties of the AABW varieties?

P14, first paragraph: again, please give the values. What is the "very shallow depth"? What do you mean l10 by "a much smaller scale"? Give depth / volume values.

P14: so you seem to say that the polynya opening in SoSE is the consequence of WDW upwelling. What is causing it? You don't need to formally prove it, but check if someone has looked at that already or suggest potential mechanisms.

P15, l1: this comment is the one reason I waived my anonymity already in the previous round of reviews: No, this is not what Heuzé et al. (2013) says! **As said during the previous review**, please change to Kjellsson et al. (2015) or Heuzé et al. (2015). Note that your author response document says you did change it to Heuzé et al. (2015).

P16, l1: where can that "be seen"?

P16, l6: likewise, where do you show that it is colder and with lower salinity?

P17, l17: "a higher content" than what?

P17, l31: the long-term warming of the bottom waters has been found by other people with other models (although related to your reanalyses). Cite for example Martin et al. (2012) https://link.springer.com/article/10.1007/s00382-012-1586-7 or Dufour et al. (2017) http://journals.ametsoc.org/doi/abs/10.1175/JCLI-D-16-0586.1

Figure 8: typo in caption, this figure shows ECCO2, not UR025.4

---

## Author Response (AR2)

**REPLY LETTER**

Received: 23 August 2017

In the following document, the original reviewer comments are in **Bold**. The author's responses are in plain font, and the alterations in the text are in *Italics*. Please note that a separate detailed reply for each referee is provided.

5

Referee #1: Anonymous referee

Overall I am satisfied with the revision. The authors explained the processes governing AABW formation in each product. I just have a few minor concerns.

Since it could take thousands of years to reach equilibrium conditions, even after twelve years adjustment could be in progress. I don't think "a one year spin-up procedure" would lead to equilibrium conditions. The weakening of vertical stratification that induces polynya could be a part of adjustment. There is nothing the authors can do concerning the adjustment, but I don't think ignoring it is not a proper interpretation.

We would like to thank the referee again for taking the time to review the manuscript, and also for the minor suggestions. We agree that addressing the small or lack of spin-up period in the reanalysis products are necessary, since it could be one of the issues that lead to polynya opening in both ECCO2 and UR025.4. In order to clarify 20 that. the following statement was added in the summary section of the manuscript:

"In this matter, the low spin-up period of 1 year in SoSE and lack of spin up in ECCO2 could have allowed instability in water column in the reanalysis, hence weakening stratification."-Page 20, Lines 4 to 6.

25

15

**Lines 9-12 I am not sure what the authors are trying to convey. Are you trying to say that even in data assimilated products the error is not fixed?**

Yes. We have also added the assimilatory adjective to the statement, in order to make the idea clear. The new text

30 is:

"Although those events are well described in non-assimilatory ocean simulations, the recent appearance of a massive open-ocean polynya in the Estimating the Circulation and Climate of the Ocean Phase II reanalysis product (ECCO2) raises questions on which mechanisms are responsible for those spurious events and if they are also present in other state-of-the-art assimilatory reanalysis products." - Page 1, Line 9-12.

35

Thank you again for the suggestions.

The manuscript has gained a lot in clarity, but part of the narrative is still confusing and does not seem to

- 5 hold. In particular, from section 3.1 to 3.2, then 3.2 to 3.3, the years where the described events start seem to change. I suggest changes in this document in order to clarify the chronology of the events and verify that the proposed mechanisms are valid.
- 10 We would like to thank again the referee for taking the time to review the manuscript in such a detailed manner, and also for all the specific comments. We believe that the valuable review and suggestions have translated into a great enhancement on the manuscript quality. We have noted that the dates of the anomalous events in each sections are different, but that is because the timing of the event in the Sea Ice, Temperature, Salinity and Neutral density is not necessarily the same. For example, when discussing the signature in sea ice averages, the SIC of
- 15 ECCO2 in Weddell Sea starts to decrease in 2004, even though the polynya had open in November 2003. That is because the sea ice values are annual averages, and since the polynya only appeared in Weddell Sea at the end of 2003; thus the decrease in SIC during this year is not enough to diminish spatially and annually averaged SIC values. In Section 3.3 we highlight that the cooling and freshening of the bottom layer of each sector of ECCO2 happens after 2006, even though the Weddell Polynya opened in November 2003. But that again is because the
- 20 temperature and salinity anomaly trends are averages for each cell of the model contained within the layer described in the Methods section. Hence, the trends in temperature and salinity only appear in the monthly averages after the newly formed AABW had substantially spread through the bottom layer, i.e. years later. We agree that the lack of this information can create confusion to the reader, and hence the information was added to the text.
- 25

For the ECCO2, was added:

"It is important to highlight that the timing of the signals in SIC, SIT, temperature, salinity and neutral density are different. First, even though the polynya appearing in ECCO2 opens in November 2003, the signal of decreasing SIC and SIT appear only from 2004 onwards. That is because the sea ice data used in this study are annual averages, and since the polynya only opened at the end of 2003, its signal was not enough to diminish the SIC and SIT annual averages. Also, even though the polynya was already established in Weddell Sea in 2004, the freshening and cooling signals in the bottom layer of ECCO2 are noticeable only after 2006 (Figure 8c and 8f). Again, that is because the monthly average temperatures and salinities were calculated, for each cell, as a mean weighted by the volume of the cell. Hence, the signals in temperature and salinity only appear in the bottom layer

- after the new volume of the bottom water has been replaced. Finally, it is also important to note that, even though the Polynya had opened in November 2003, the bottom water production (WSDW and WSBW) signal appeared in the intermediate layer in 2007 onwards (Figure 11e)." – Page 13, Lines 30 to 33 and Page 14, Lines 1 to 6.
- 40 For the SoSE,

"The timing of the events in SoSE are more tied together, but that is because as soon as the polynya opens, WSDW is formed and transported to the bottom layer (Figure 12c), thus having minimum lag between the ice-free area opening and the WSDW formation. Hence, we can see prior and during the polynya opening a clear warming of the bottom layer and cooling of intermediate layer, which weakened vertical stratification and allowed WSDW to be

45 the bottom layer and cooling of intermediate layer, transported downwards." – Page 15, Lines 1 to 4.

And for the UR025.4

"With respect to timing, signals in sea ice variables and ocean variables in UR025.4 had minimum lag. That is because the AABW is formed here rather abruptly. SIT annual averages only reached its peak in Weddell Sea in 2009, even though the AABW formation in this sector occurred in the winter of 2008. However, the sea ice increase in the annual average is noticeable from 2006 until 2009, showing that the major sea ice production happened

- 5 during this period, which is in agreement with the AABW formation in 2008 at Weddell Sea Sector by brine release. Regarding the salinity anomaly signals in the Indian Ocean Sector, it is possible to see that the minimum salinities happened almost simultaneously in the intermediate and surface level (Figure 9 d and e). That is because the UCDW entrainment responsible for this fresh signal happened simultaneously in the intermediate and surface layer. Moreover, the following high salinity signal in all three layers are simultaneous (Figure 9 d-f), as the newly formed
- 10 bottom water is swiftly injected to the bottom layer (Supplementary 2). Finally, pulses of WSDW production were described in Weddell Sea sector from 2005 until 2008, however their magnitude were too small to print signatures in temperature and salinity mean values." Page 18, lines 19 to 29.

**Major changes**

**15 For each reanalysis, please indicate**

**- In the text at the beginning of section 3.2, the dates of the polynyas given in section 3.1;**

Thank you for the suggestion. As requested, in the paragraphs which we discuss each reanalysis, we added the timing of the polynya opening. For ECCO2, the sentence in page 8 line 9 and 10 was written as "The WDW is warmer and saltier than AASW, so intensive winter surface cooling of this water mass has the potential to form

20 warmer and saltier than AASW, so intensive winter surface cooling of this water mass has the potential to form WSDW and even WSBW through open ocean convection". The sentence now is:

"The WDW is warmer and saltier than AASW, so intensive winter surface cooling of this water mass due to polynya opening from November of 2003 until the end of the reanalysis period has the potential to form WSDW and even WSBW through open ocean convection." – Page 8, Lines 13 to 15.

**And for the SoSE**

"From May until November of 2005, i.e. while an open ocean polynya is open in Weddell Sea, the relatively highwater volume of 32.6% of WDW swiftly decreases to 26% (Figure 3 - WDW)." – Page 8, Lines 32 and 33.

30

25

**- On figures 3 to 7, these same dates should be represented as vertical bars with different colours and/or line styles, for example.**

Thank you for the suggestion. A green vertical shading bar was added to those figures to highlight the period in which the polynyas stays open in SoSE, and a yellow shading bar for when ECCO2 polynya is open. Figures 3-7 are now located from page 28 to 32.

**- In the text at the beginning of section 3.3, the dates at which AABW formation occurs in each sea inferred from figures 3 to 7;**

40

Thank you for the suggestion. The following addition was done in the first paragraph in order to cite the AABW formation dates:

"The investigation of the temperature and salinity time series focus on the periods and locations of identified AABW
formation in each reanalysis output, which in ECCO2 was from 2004 until 2012 at Weddell Sea, Indian Ocean and Western Pacific Sectors; in SoSE at the Weddell Sea in 2005; and in UR025.4 at the Indian Ocean and Western Pacific Sectors in 2004 and Weddell Sea Sector in 2008." – Page 10, Lines 18 to 21;

- On figures 8 to 10, again potentially as vertical bars or shaded rectangles of different colours for the different regions.

Thank you for the suggestions. Vertical bars were added for the periods of AABW formation in the three reanalysis. Figures 8-10 are now from page 33 to 35.

Then throughout the text, check that you do describe what happens at those dates that are specific to each 5 region and each reanalysis. For example page 11 line 18, you explain that the Weddell Sea sector does not show the same changes as the other two regions in 2004 in UR025.4. That is not surprising, since as you write page 8 lines 22-23, there WSBW is formed only in 2008 and 2009! Likewise, page 12 you write that WSBW formation occurs "in the first few months of the time series" for SoSE, yet your description page 8

mentions WBSW pulses later, in 2006 and 2007. 10

Thank you again for the suggestion. We agree that is necessary to also cite the dates of AABW production in SoSE after 2005. So the sentence was edited to add the information requested:

15 "Finally, in SoSE, the WSBW and WSDW formation occur mostly during the first few months of the time series in 2005, with smaller total formation pulses (WSBW +WSDW) in 2006 and 2007 also (Figure 3- WSDW and WSBW)." - Page 12, Lines 19 to 20;

We also agree that the sentence

"The Weddell Sea sector does not show either of these strong freshening or salinity increasing patterns in the 20 intermediate laver."

on page 11. line 18 was redundant, so the sentence was removed from the manuscript.

**Minor changes**

**25 Throughout the text: despite the language check, there are a lot of typos, blank spaces or letters missing.**

Thank you again. The typos and blank spaces were checked.

**P1, I9: Change "despite" into "although".**

30

The suggested change was done.

**P1, I23: "longer and more elegant" than what?**

35 We agree that the use of the comparative words are incorrect. We instead changed the words to adjectives since we are not using comparison:

"Recently, different groups of experts have developed several state-of-the-art eddy-permitting general ocean circulation models with long simulations, elegant and efficient assimilation methods." - Page 1, Lines 23 to 24;

40

**P3: Potentially split section 2 into two subsections, one about "Reanalysis and observational products" starting line 10, and one about "Methods" starting P5 I7.**

We have split the section as proposed. Thank you for the suggestion. Section 2.1 (Ocean Reanalysis Datasets and 45 Observations) starts now on page 3, line 10. Section 2.2 (Methods) starts now in Page 5, Line 8.

P4, I25-26: This sentence has no clear link with the one after. Move it somewhere else, or add some transition.

This sentence was supposed to be located in the end of the previous paragraph, after describing the difference in the simulation frameworks of the three reanalysis products. We agree that it was misplaced during the reviewing process, and hence the phrase was put back in the end of the paragraph, on lines 24 to 26 of page 4.

5

**P5, I32: "their links with the processes being evaluated" – reference needed.**

Thank you again. As requested the reference used was added:

10 "The depth limits of the three layers were chosen specifically due to their links with the processes being evaluated (Orsi et al., 1999)." – Page 6, Lines 1 and 2.

**P7, I14: mostly in the Weddell Sea.**

15 Thank you for the suggestion. We added the location reference on the sentence:

"Although no observational SIT database that efficiently covers the Southern Ocean is currently available to our knowledge, the comparisons among the three reanalysis thicknesses and the magnitudes of the signals suggest an UR025.4 overestimate, especially in Weddell Sea." – Page 7, Lines 16 to 19.

20

**P8, I12 and I20: this comment is valid throughout the manuscript and was already made during the previous round of reviews: give the actual values! Don't just write that it increases**

Thank you for the suggestion. As requested values were added throughout the text. It follows the detailed description of the additions.

- Page 8, previous Lines 13 where it was written "volume percentages increase sharply.", now is written "volume percentages increase sharply in 10% during the following 4 years." placed now in line 17.

30 - Page 8, previous lines 13 and 14 where it was written "During the WSBW formation, WSDW is no longer formed, and rapid conversion of WSDW to WSBW occurs.", now is written "During the WSBW formation, WSDW is no longer formed, and rapid conversion of 42% of WSDW to WSBW occurs." placed now in lines 18-19.

Page 8, previous lines 18 and 19, it was written "Thus, the total volume of AASW declines throughout the time series", now it is "*Thus, the total volume of AASW declines in total 5.5% throughout the time series.*" now placed in lines 23 to 24.

Page 8, previous lines 19 and 20, it was written "During consecutive winters, the WDW volumes drop, whereas the percentages of WSDW and WSBW slightly increase." now is written "During consecutive winters, the WDW volumes drop by 2 to 6%, whereas the percentages of WSDW and WSBW slightly increase by the same total percentage volume." now in lines 24 to 26.

**P8, I22: for clarity, write ECCO2 instead of "the previous reanalysis product".**

45 The alteration was done as suggested.

**P8, I25: the behaviour of SoSE is similar to that of ECCO2 only until 2008. Please discuss what happens from 2008 onwards.**

Thank you for the suggestion. We agree that the following paragraph was missing the information on the water mass changes after 2008. Since there is no AABW formation in SoSE after 2008, we then added the following sentence:

5 "After 2008, SoSE do not show either WSDW or WSBW formation (Figure 3 – WSDW and WSBW), while both AASW and WDW increase steadily by less than 5% from 2008 until 2010." – Page 9, Lines 7 and 8.

Also, to make clear that the similarity described happen only prior to 2008, the first sentence of the paragraph was changed to:

10

"The SoSE reanalysis product shows similar water mass alterations to that of the ECCO2 product prior to 2008." – Page 8, Line 30.

**P10, I13: see major comment, give clearly "the periods and locations".**

15

Thank you again. We agree that detailed pointing of place and timing of AABW formation in each reanalysis was necessary here. The following statements were added:

"The investigation of the temperature and salinity time series focus on the periods and locations of identified AABW
formation in each reanalysis output, which in ECCO2 is from 2004 until 2012 at Weddell Sea, Indian Ocean and Western Pacific Sectors; in SoSE is at Weddell Sea in 2005; and in UR025.4 at Indian Ocean and Western Pacific Sectors in 2004 and Weddell Sea Sector in 2008." – Page 10, Lines 18 to 21.

25 P10, I27-28: no, it is not apparent that cooling was the main mechanism. You do not show it, and you even soften your argument several lines later by writing that cooling could be "favouring deep convection". I'd remove the sentence I27-28 and keep the ones after.

Thank you again. As you suggested, we believe that the use of the word "apparent" was misplaced in this part of the text. The idea that was being transmitted was that as density increase is necessary to create AABW, and freshening experienced in ECCO2 sectors would decrease neutral densities, then cooling is likely the process by which waters gained enough density to create AABW. However, the word "apparent" gives an idea of certainty that can be misleading, and hence we changed the phrase to the following structure:

35 "Since freshening lowers water mass densities, cooling might be one mechanism responsible for AABW formation in ECCO2." – Page 11, Lines 3 and 4.

**P11, I12: what do you mean by "consistent with the changes in temperature"? That it happens simultaneously? That the increase in salinity and temperature makes sense you think (if so, explain why)?**

Yes, we mean simultaneously. For clarity, the sentence was changed to:

"The salinity anomalies in UR025.4 show alterations during the AABW formation period (2004) simultaneously with the changes in temperature. Before 2004, the bottom layer salinity appeared to decrease slowly (Figure 9d)."- Page 11, Lines 20 and 21.

**P11, I24 onwards: not proven. You need to talk more about the timing of the events, how the salinity anomaly would propagate from the surface down.**

50

Thank you for this comment. We see that the analysis of AABW formation in Prydz Bay still cast doubts. In order to further clarify the process, we have constructed two animations with SIT anomalies in Prydz Bay and Salinity anomalies too, during the year 2004, i.e. when the AABW is formed. Because the process of AABW formation in this area in UR025.4 is related to sea ice formation, then salinity can be used as a *tracer* of the dense water mass

- 5 formation, and hence positive salinity anomalies can be linked the waters being formed. In the animation, it is also included the year before the AABW formation (2003), as an example of the natural seasonal cycle in Prydz bay when AABW is not formed. Note in the animation with the sea ice thickness anomaly (Suplementary1.gif), that the depth used is close to the one on figure 13 of the main manuscript, where we show the sea ice velocities. In 2004 (Suplementary1.gif, April-2004), a region with high positive SIT anomalies appear west of Prydz bay, and the water
- 10 column under it immediately respond with a high salinity signal, from sea ice formation. The high SIT signal persists locally until July, while the high salinity anomaly is present until mid-2005. The period with the highest salinity anomalies happen between May and October of 2004, which is the period where AABW pulse is identified in the section 3.2 of the manuscript, in Indian Ocean Sector. In a second animation, we present salinity anomaly contours, also in Prydz Bay, but in three different depths (457 m, 2262 m, and 3513 m). The deepest contour belongs to the
- 15 bottom layer definition of the layers used in section 3.3. In this second animation (Suplementary2.gif) it is possible to see that in March the region with high salinity anomaly appears in the intermediate layer. In the following month (April) we can already see that this signal had propagated to the 2262 m layer, and in May it reached the deepest salinity anomaly contour. In the following months (July-2004 until October-2004), we can clearly see the high salinity anomaly moving downwards from the intermediate layer, and spreading laterally, showing that the dense water
- 20 formed due sea ice formation reached density values enough to be transported to the bottom layer. Even after October 2004, the plume of high salinity persisted in the bottom layer, expanding laterally, showing the lateral advection of this dense saline water. Now, the period of downslope flow of this high salinity water coincides with the period of AABW formation in Indian Ocean Sector in section 3.2, showing that the salinity increase due to brine release was likely the process responsible to raise the neutral density of coastal waters enough to create AABW,
- 25 and transport it downslope until the bottom layer. The period from October-2004 to Jan-2005, when high salinity waters in the 3513 m layer are still expanding, there was no AABW formation according to the volumes calculated through neutral densities (section 3.2). Hence, we believe the high salinity AABW is only being horizontally advected along that layer.
- 30 Relative to the time scales, as can be seen in the animation, one month after the salinity anomalies appear in the intermediate layer, it appears in the top of the bottom layer too. Hence, as the reanalysis has monthly averaged output, it is not possible to calculate accurately the timing of the propagation of salinity anomalies to the bottom layer due to lack of temporal resolution.
- 35 We added also the following animations as supporting material to the article, in order to make this explanation available to anybody that has access to the manuscript. Also, the supplementary material content is referenced on the text:

"In fact, neutral density contours along Prydz Bay show salinity anomalies increasing and being exported to the bottom layer as SIT anomalies grow (Supplementary material)." – Page 12, Lines 4 to 6.

40

**P13, I4-6: where do you show the properties of the AABW varieties?**

45 Thank you for the comment. In lines 4 to 6, we are specifically addressing the cooling and freshening noticed in the bottom layer properties of ECCO2. Since the bottom layer of the Weddell Sea sector, as divided in the manuscript, is mainly comprised by waters of AABW, the cooling and freshening described in Weddell Sea bottom layer in the section 3.3 translates into a cooling and freshening of AABW. However, it seems that the sentence as described did not properly explained the idea, and hence, that was changed from

"The temperature and salinity decreases described in section 3.3 in the bottom layer of ECCO2 after 2006 are related to the polynya appearance. The AABW varieties formed under the Weddell polynya retain the distinct low salinity and low temperature signals due to heat loss at the surface." - Page 13, Lines 4 to 6.

- 5 to "The temperature and salinity decreases described in section 3.3 in the bottom layer of ECCO2 after 2006 are related to the polynya appearance and AABW formation, since the bottom layer of the Weddell Sea in this reanalysis essentially contains AABW. Hence, since cooling and freshening is present in bottom layer of Weddell Sea sector in ECCO2, we suggest that AABW varieties formed under the Weddell polynya retain the distinct low salinity and low temperature signals due to heat loss at the surface." Page 13, Lines 13 to 17.
- 10

**P14, first paragraph: again, please give the values. What is the "very shallow depth"? What do you mean 110 by "a much smaller scale"? Give depth / volume values.**

Thank you again. The suggestions were added as following:

15

-In page 14, lines 2 and 3, it was added the depth requested (10 m). The phrase now is written as "*From January to May, before the polynya opens, the presence of WDW at 10 m at approximately 70°W is noticeable in the SoSE neutral density transects.*" in lines 25 and 26.

20 -In page 14, lines 9 and 10, we also added the volume change described by the smaller scale. Now the sentence is written as "*WSDW formation also occurs during the following two winters, but with volumes less than half of the 6% production in 2005.*" in lines 32 to 33.

**P14: so you seem to say that the polynya opening in SoSE is the consequence of WDW upwelling. What is causing it? You don't need to formally prove it, but check if someone has looked at that already or suggest potential mechanisms.**

Thank you for the suggestion. We agree that is necessary to point out which processes could be leading to WDW uplift in the reanalysis. Hence, the following paragraph was added to the text:

30

"It seems that WDW uprising is the main mechanism responsible for melting the sea ice, and creating the Weddell polynya in both ECCO2 and SoSE. Although out of the scope of this study, some processes can cause isopycnal uplift in Weddell Sea, creating the open ocean polynya. An experiment with global ocean-sea ice model performed by Hirabara et al., (2012) has suggested that a saline surface layer and persistent cyclonic wind stress are

- 35 necessary to lower vertical stratification and allow WDW uplift over the Maud Rise. In a recent attempt to reproduce the Weddell Polynya, Cheon et al. (2015) has found that the establishment of strong negative wind stress curl in Weddell Sea accelerates the Weddell gyre, causing WDW to upwell in the center of the gyre and melting sea ice. Furthermore, a simulation with the Kiel Climate Model (KCM) has shown that warm waters built-up in Weddell Sea deep layer during non-convective periods, and after decades the heat buffered interact with sea ice opening the Weddell Polynya (Martin et al., 2013)." – Page 15. Lines 15 to 23.
  - P15, I1: this comment is the one reason I waived my anonymity already in the previous round of reviews: No, this is not what Heuzé et al. (2013) says! As said during the previous review, please change to Kjellsson et al. (2015) or Heuzé et al. (2015). Note that your author response document says you did change it to

45 Heuzé et al. (2015).

You are right. Sorry for the typo. As requested, the reference was changed to Heuzé et al. (2015).

P16, I1: where can that "be seen"?

The use of "be seen" is also misplaced. We changed for "hence". Now the phrase reads: "Hence, UR025.4 rather accurately represents both the warm water entrance into Prvdz Bay and the density increase along the circulation present in the real world." – Page 17, Lines 1 to 3.

5

**P16. I6: likewise, where do you show that it is colder and with lower salinity?**

Thank you for the comment. We analyzed maps of temperature and salinity anomaly throughout the time series. and through the maps it was possible to see the low temperature and salinity anomalies propagating from Indian

10 Ocean Sector towards Weddell Sea Sector, However, those maps were not shown in the final manuscript. We hence changed the sentences to:

"Thoroughly inspecting monthly maps of salinity and temperature anomalies (not shown) revealed that UCDW originating from the Indian Ocean entering the Weddell Sea in UR025.4 is colder and with lower salinity than the local WDW in the Weddell Sea, especially due to the melting episode that occurred in Prydz Bay in early 2004." -

15 Page 17, lines 7 to 9.

**P17, I17: "a higher content" than what?**

20 Thank you. We agree this sentence is confusing. For clarity, we changed to:

"First, the real AABW formation in the Indian Ocean sector occurs after the modified Circumpolar Deep Water (mCDW) circulates deeper under the ice shelves surrounding Prvdz and Vincennes Bays, mixing with the DSW created in the coastal polynyas and increasing its salt content as well as lowering its temperatures (Williams et al.,

25 2016)." - Page 18, Lines 7 to 10.

> P17, I31: the long-term warming of the bottom waters has been found by other people with other models related Cite (although to your reanalyses). for example Martin et al. (2012)https://link.springer.com/article/10.1007/s00382-012-1586-7 or Dufour et al. (2017)

30 http://journals.ametsoc.org/doi/abs/10.1175/JCLI-D-16-0586.1

Thank you for the suggestions. As requested, we added the reference of both studies mentioned:

"Long term warming of bottom waters have been pointed as a trend also in other non-assimilatory models, such as the Kiel Climate Model (Martin et al., 2013) and Climate models CM2.5 and CM2.6 (Dufour et al., 2017), and in 35 both studies the heat buffered has opened a polynya in Weddell Sea." - Page 19, Lines 11 to 13.

**Figure 8: typo in caption, this figure shows ECCO2, not UR025.4**

Thank you for noticing. The typo was fixed. 40

---

## Author Response (AR4)

**REPLY LETTER FOR THE EDITOR**

Date of reply: October 27, 2017

The last changes made by the authors, relative to the submitted version of the manuscript of number os-2017-9 are described in details. Sentences from the manuscript are in *Italics*. Additions within the manuscript are also **bold**.
* * *
During the typesetting process, some sentences and references were added to the text. This reply letter describe those changes, in order to make those changes available for the scientific community.  All the authors would like to thank the editor and typesetters for the opportunity to describe the changes done in the manuscript. It follows a list of changes done in the manuscript:

→Page 2, Lines 8-10 – It came to our attention that a reference is missing here. The appropriate reference was added:
*"The dense bottom waters in the Southern Ocean are mainly formed by two mechanisms. The first mechanism is through (i) a complex interaction of deep and shelf waters and starts with deep waters, originally formed in the North Atlantic, being transported to the south through the AMOC **(Ferreira and Kerr, 2017).***"

→Page 2, Lines 27-29 - Recently, a new open ocean polynya in the prime meridian has appeared (2016 and 2017), and we believe that not mentioning that polynya in this phrase might convey an erroneous information.  The sentence was then changed from:
"*Although smaller ocean polynyas occurred in the 20th century (Comiso and Gordon, 1987), no winter ice-free areas in the Southern Ocean with the dimensions and persistence of the Weddell Polynya have been reported since the 1970s.*"

To the following:
"*Although smaller ocean polynyas occurred in the 20th century (Comiso and Gordon, 1987) **and in 2016 and 2017**, no winter ice-free areas in the Southern Ocean with the dimensions and persistence of the Weddell Polynya have been reported since the 1970s*"

→Page 3, Lines 21-23 – The correct date of the reference is 2008. The text now is:
*"A Green's function method is used to calibrate the control variables (Menemenlis et al., **2008**) and the initial parameters, which include initial temperature and salinity condition…"*

→Page 4, Lines 22-23 – The communication year was missing. The text now is:
*"Furthermore, neither the OCCA nor the SoSE were optimized to eliminate spurious drifts (M. Mazloff, personal communication, **2017**)."*

→Page 6, Lines 20-21 – Plural is missing. The phrase now is:
*"In this section, we first describe the average sea ice patterns in the Southern Ocean **sectors**, its spatial signature and evidence that this property is related to the AABW formation in the reanalysis products investigated (Sect. 3.1).*

→Page 6, Lines 21-23 – The information in the parenthesis is redundant, and we chose to remove from the phrase. The sentence was changed from:
*"Section 3.2 discusses the water mass volume transformations in each sector, and attempts to identify the AABW formation in the different products (Sec. 3.2)."*

To the following:
*"Section 3.2 discusses the water mass volume transformations in each sector, and attempts to identify the AABW formation in the different products."*

→Page 6, Lines 28-30 – The sector name is incomplete. The phrase was changed to:
*"All three reanalysis products overestimate the annual mean SIC compared to the National Snow and Ice Data Center observations (hereafter referred to as NSIDC) in all sectors until 2004, except in the Ross and Bellingshausen **and Amundsen**, where the concentrations are similar to the observations obtained from the NSIDC (Figure 2a-f)."*

→Page 8, Lines 12-13 – The preposition is miswritten, and changed from *in* to *after*. The sentence now is:
*"This change leads the decrease of WDW to volume percentages lower than 10% **after** 2012 (Figure 3 – WDW)."*

→ Page 10, Lines 15-18. It came to our attention that one important reference used that describe oceanographic processes in the Bellingshausen and Amundsen sector is missing. Hence the following reference in Bold was added:
*"Finally, no pulse of AABW is seen in any of the three reanalysis in the Bellingshausen and Amundsen sector (Figure 7 - AABW), which is expected since this sector in the Southern Ocean lacks hydrographic, shelf morphology and cryosphere conditions required to create AABW varieties (Potter and Paren, 1985; **Whitworth III et al., 1998;** Orsi et al., 1999)."*

→Page 11, Line 28-30. The word *into* was misplaced, and changed to *from.* The sentence now is:
*"The bottom layers of both the Western Pacific and Indian Ocean sectors show an increase in salinity between May and September of 2004, which denotes a downward propagation of saline waters **from** the intermediate layer (Figure 9f)."*

→Page 12, Line 15-16. The word *Finally* is repeated. This word was deleted. The full sentence now is:
*"The second alteration evident in the UR05.4 anomalies was a warming of the Western Pacific and Indian Ocean sector bottom layers between May and October of 2004 (Figure 9c)."*

→Page 13, Lines 2-4. The preposition in was missing. Now the sentence is:
*"The anomalous signals identified by the average SIC and SIT distribution in ECCO2 are mainly connected to the appearance of a large-scale sensible heat polynya **in** the Weddell Sea sector (Figure 11a-c) and the neutral density alterations (Figure 11d-f), as previously pointed out by Azaneu et al. (2014)."*

→Page 13, Lines 6-8. The word monthly was miswritten, since we analyzed annual means. Now the sentence is:

*"This low sea ice content signal is extreme enough to decrease the **annual** mean sea ice concentrations and thicknesses in the Weddell Sea and Indian Ocean sectors, and even the whole Southern Ocean average (Figure 2a,b,e)."*

→Page 13, Lines 25-28. The abbreviation eOMP was no longer used in the text, so it was removed:

*"Pardo et al., (2012) used extended Optimum Multiparameter Analysis to quantify the volumes of the Southern Ocean water masses and found that the longitudinal limits of our Weddell Sea sector was filled with approximately 26±0.2% of WSBW, a percentage substantially lower than the 70% of WSBW estimated by ECCO2 in 2013."*

→Page 15, Lines 21-23. The abbreviation KCM was no longer used in the text, so it was removed:

*"Furthermore, a simulation with the Kiel Climate Model have shown that warm waters built-up in Weddell Sea deep layer during non-convective periods, and after decades the heat buffered interact with sea ice opening the Weddell Polynya (Martin et al., 2013)."*

→Page 16, Lines 9-12. The communication year that was missing is now added:

*"Finally, because the polynya in SoSE occurs at the beginning of the reanalysis output, we cannot assure its opening is a result of an initial adjustment process, even though a one year spin-up procedure is conducted in the prior year (2004) to bring the SoSE to its equilibrium conditions (M. Mazloff, personal communication, **2017**)."*

→Page 18, Lines 28-29. The word *Finally* was repeated, and is removed from the sentence in this version of the manuscript:

*"Pulses of WSDW production were described in Weddell Sea sector from 2005 until 2008, however their magnitude were too small to print signatures in temperature and salinity mean values."*

→Page 19, Lines 21-23. The word *propagation* was misused here, and it was changed to *appearance* in this version of the manuscript:

*"The **appearance** of spurious open ocean deep convection in the Southern Ocean simulations can go even further and cause the warming of the abyssal layer, the cooling of surface waters and atmospheric warming (Latif et al. 2013; Pedro et al. 2016)."*

→Page 19, Lines 26-29. The reference year is wrong, and was corrected to 2014:

*"Especially in ECCO2, the mechanism of AABW formation resulted in erroneous representations of the Southern Ocean, such as high AABW volumes and lower sea ice concentrations and thicknesses, reinforcing that open ocean deep convection inserts errors in the simulation (e.g., Azaneu et al., **2014**; this study)."*

➔Page 20, Lines 5-7. The word *consistent* was changed to *slight* in this sentence.
*"However, the WDW increase reported here is consistent with the observed results reported by Kerr et al. (2017 - under review), who found a **slight** increase of the WDW contribution to the total mixture of deep and bottom waters in the Weddell Sea from 1984 to 2014, despite the high degree of interannual variability."*

➔Page 20, Lines 8-10. The phrase was changed from:
*"However, since no real open ocean polynya has been reported since the 1970s (Gordon 1978), a critical analysis of the model mechanisms of heat exchange between the surface waters and sea ice is required in the future to efficiently understand the role of WDW in open ocean polynya establishment."*

To the following:
*"However, since no real open ocean polynya has been reported for this period, a critical analysis of the model mechanisms of heat exchange between the surface waters and sea ice is required in the future to efficiently understand the role of WDW in open ocean polynya establishment."*

➔Page 20, Lines 10-12. A missing reference was added.
*"In addition, since bottom layer warming and intermediate layer cooling are the possible mechanisms that diminished stratification in ECCO2, further evaluation of the causes of those trends is needed to understand the primary factors leading to the weak ocean surface stratification **(Azaneu et al., 2013)**."*

➔Page 20, Lines 12-14 - Again, the authors see that is necessary to point out the recent polynya appearance in 2016 and 2017. This occurrence is strictly linked to the subject of this study, and cannot be ignored in the paper. Hence, the authors have added the following phrase:
*"Finally, the appearance of a real open ocean polynya in October 2016 and September 2017 on Weddell Sea will also furnish a study scenario to understand heat exchange between sea ice and warmer waters, and compare with ocean reanalysis products."*